# Ribosome profiling at isoform level reveals evolutionary conserved impacts of differential splicing on the proteome

Marina Reixachs-Solé[1,2], Jorge Ruiz-Orera[3], M. Mar Albà [4,5,6] & Eduardo Eyras [1,2,4,5✉]

The differential production of transcript isoforms from gene loci is a key cellular mechanism. Yet, its impact in protein production remains an open question. Here, we describe ORQAS (ORF quantification pipeline for alternative splicing), a pipeline for the translation quantification of individual transcript isoforms using ribosome-protected mRNA fragments (ribosome profiling). We find evidence of translation for 40–50% of the expressed isoforms in human and mouse, with 53% of the expressed genes having more than one translated isoform in human, and 33% in mouse. Differential splicing analysis revealed that about 40% of the splicing changes at RNA level are concordant with changes in translation. Furthermore, orthologous cassette exons between human and mouse preserve the directionality of the change, and are enriched in microexons in a comparison between glia and glioma. ORQAS leverages ribosome profiling to uncover a widespread and evolutionarily conserved impact of differential splicing on translation, particularly of microexon-containing isoforms.

[1] The John Curtin School of Medical Research, Australian National University, Canberra, ACT 2601, Australia. [2] EMBL Australia Partner Laboratory Network at the Australian National University, Canberra, ACT 2601, Australia. [3] Max Delbrück Center for Molecular Medicine in the Helmholtz Association, Berlin 13125, Germany. [4] IMIM - Hospital del Mar Medical Research Institute, E08003 Barcelona, Spain. [5] Catalan Institution for Research and Advanced Studies, E08010 Barcelona, Spain. [6] Pompeu Fabra University, E08003 Barcelona, Spain. ✉email: eduardo.eyras@anu.edu.au

The alternative processing of transcribed genomic loci through transcript initiation, splicing, and polyadenylation determines the repertoire of RNA molecules in cells[1]. Differential production of transcript isoforms, especially through the mechanism of alternative splicing, is crucial in multiple biological processes such as cell differentiation, acquisition of tissue-specific functions, and DNA repair[2–4], as well as in multiple pathologies[5–7]. Although analysis of RNA-sequencing (RNA-seq) data from multiple samples has indicated a large diversity of transcript molecules[8], genes express mostly one single isoform in any given condition and this isoform may change across conditions[9,10].

Computational and in vitro studies have provided evidence that a change in relative isoform abundances can lead to the production of protein variants that impact the network of protein–protein interactions in different contexts[11–14]. In contrast, quantitative proteomics of naturally occurring proteins has identified much fewer protein variants than those predicted with RNA sequencing[15,16]. Using state-of-the-art proteomics, it was recently shown that splicing changes at the RNA level lead to changes in the sequence and abundance of proteins produced, although this was detected only for a limited number of transcripts[16]. The difficulty in establishing a correspondence between transcript and protein variation may be not only due to limitations in current proteomics technologies, but also to the stability and translation regulation of transcripts[17,18]. Despite the evidence about its functional relevance[3], it is still debated whether differential splicing leads to fundamentally different proteins and how widespread this might be[19–21]. Of particular interest are microexons, which can be as short as three nucleotides and carry out conserved neuronal-specific functions, and whose misregulation is linked to autism[22–24]. Despite their involvement in protein–protein interactions[23,25], the detection of protein variation associated with differential microexon inclusion using proteomics is currently challenging.

Sequencing of ribosome-protected RNA fragments, i.e., ribosome profiling, provides information on the messengers being translated in a cell. In particular, it allows the identification of multiple translated open-reading frames (ORFs) in the same gene and the discovery of novel translated genes[26–29]. However, ribosome profiling studies have been mainly oriented to gene-level analysis[26,28,30]. Recently, reads from ribosome profiling have been mapped across the exon–exon junctions of alternative splicing events[31], suggesting that alternative splicing products may be engaged by ribosomes and potentially translated to produce different protein isoforms. A potential limitation of that approach is that ribosomal profiling reads also contain signals from native, non-ribosomal RNA–protein complexes[32]; hence, the mapping of ribosome reads to these regions may not necessarily be indicative of active translation. In addition, ribosome activity is associated with signal periodicity and uniformity along open-reading frames[33], which has not yet been tested in relation to transcript isoforms and alternative splicing. Thus, the extent to which alternative splicing, and in particular microexon inclusion, leads to the translation of alternative ORFs remains largely unknown.

In this paper, we describe ORQAS (ORF quantification pipeline for alternative splicing), a method to quantify translation abundance at individual transcript level from ribosome profiling by taking into account ribosome signal periodicity and uniformity per isoform. We validate the translation quantification of isoforms using independent data from polysomal fractions and proteomics. We further find a concordance between differential splicing and translation and obtain evidence for the differential translation of microexons that is conserved between human and mouse. ORQAS provides a powerful strategy to study the impacts of differential RNA processing in translation.

## Results

**Translation abundance estimation at isoform level with ORQAS.** We developed ORQAS (ORF quantification pipeline for alternative splicing) for the estimation of isoform-specific translation abundance and to investigate the impact of differential splicing on translation (Fig. 1a) (see the "Methods" section). ORQAS quantifies the abundance of open-reading frames (ORFs) in RNA space from RNA sequencing (RNA-seq) in transcript per million (TPM) units, and assigns ribosome-sequencing (Ribo-seq) reads to the same ORFs using RiboMap[34]. After the assignment of Ribo-seq reads to isoform-specific ORFs, ORQAS only considers ORFs with at least ten Ribo-seq reads after pooling replicates, and with average RNA-seq abundance greater than 0.1 in TPM units. ORQAS then calculates for each of these ORFs two essential metrics to determine their potential translation: uniformity, calculated as a proportion of the maximum entropy of the read distribution, and the 3nt periodicity along the ORF. The translation abundance of each ORF is then calculated in ORF per million (OPM) units. These abundances are then used to study the impact of differential splicing on translation (see the "Methods" section).

We used ORQAS to analyze Ribo-seq and matched RNA-seq data from human and mouse glia and glioma[30], mouse hippocampus[35], and mouse embryonic stem cells[36] (Supplementary Table 1). To determine which values of uniformity and periodicity are indicative of an isoform being translated, we selected as positive controls 929 human genes with a single annotated ORF and with evidence of protein expression from mass spectrometry (MS), immunohistochemistry (IHC), and Uniprot in all 37 tissues available in the Human Protein Atlas (THPA)[37]. For the mouse samples, the positive controls were 802 genes with one-to-one orthology with the human positive controls. We considered the translation of those ORFs within 90% of the periodicity and uniformity distribution of these positive controls (Fig. 1b, Supplementary Fig. 1). This produced a total of 20,709–20,785 translated ORFs in human, and 13,019–17,515 in mouse (Supplementary Table 2).

To determine the robustness of our filter on RNA abundance, we considered other cutoffs, but the results did not change significantly after removing cases below 1 TPM (Supplementary Fig. 2). As a further quality control, we considered the proportion of isoforms with low or no RNA expression that fell inside our periodicity and uniformity cutoffs and found only 0.7–0.9% across the human samples and 0.1–1.5% in the mouse samples (Supplementary Table 3). To show that ORQAS provides an advantage over simply quantifying isoforms from Ribo-seq, we analyzed 1005 genes with one single annotated ORF not included in the list of positive controls used above, and calculated the proportion of cases that showed evidence of protein expression from IHC experiments from THPA. We observed that cases that did not pass ORQAS thresholds generally lacked protein evidence (Fig. 1c). A similar analysis with all genome-wide predictions, not including any single-ORF gene, also showed that genes with translated isoforms are more frequently validated at all levels of protein expression, and that the majority (96%) of genes with translated ORFs showed some evidence of protein expression from MS, IHC, or Uniprot (Supplementary Fig. 3a, b). To further validate the selection of these cutoffs, we considered genes with specific protein expression in the brain, heart, intestine, liver, spleen, or testis. We found that ORQAS predicts in glia a higher proportion of translated ORFs in the subset of brain-specific genes compared with the other tissues (Supplementary Fig. 3c).

ORQAS predicted that a large fraction of the expressed protein-coding genes had multiple translated isoforms: 52.3–54.9% of the genes in human and 29.1–35.9% in mouse (Supplementary Fig. 4). Overall, the majority of translated

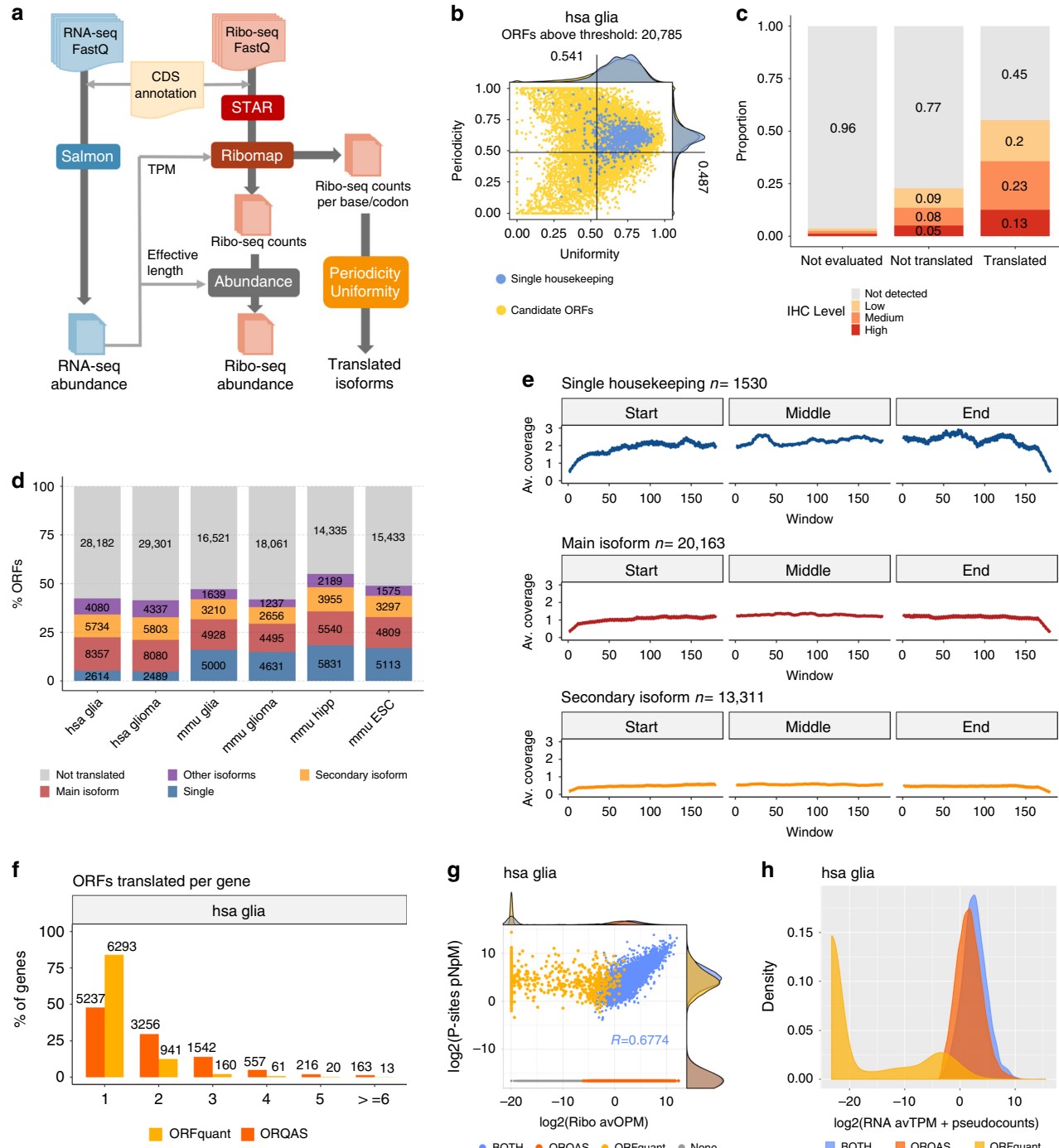

**Fig. 1 Estimation of translated isoforms. a** Diagram of the ORQAS pipeline. **b** Uniformity (x-axis) and periodicity (y-axis) for all tested ORFs with RNA expression TPM >0.1 and ≥10 Ribo-seq reads assigned (yellow) from human glia (n = 28,427). Uniformity is the percentage of maximum entropy, and periodicity is measured in the first annotated frame. In blue, single-ORF genes with protein expression in 37 tissues from the Human Protein Atlas (THPA). Other samples are shown in Supplementary Fig. 1. **c** ORQAS predictions in human glia for 1005 single-ORF genes. The plot shows the protein expression from immunohistochemistry (IHC) experiments in human cortex from THPA for cases without enough RNA expression (Not evaluated n = 468), cases that do not pass the filters of uniformity and periodicity (Not-translated n = 176), and those predicted to be translated (n = 361). **d** Number of ORFs translated per sample, according to whether the ORF is encoded by a single-ORF gene (Single), the most abundant isoform according to RNA-seq (Main isoform), the second most abundant (Secondary isoform), or by any of the remaining isoforms (Other isoforms). Tested ORFs that are not-translated are depicted in gray (Not-translated). **e** Average density of Ribo-seq reads along ORFs in the single-ORF genes from above, in ORFs from the most abundant isoform according to RNA-seq, and in the second most abundant isoform. **f** Number of ORFs translated per gene in human glia (hsa glia) according to ORQAS (orange) and ORFquant (yellow) predictions. Other samples are shown in Supplementary Fig 6. **g** Correlation between average ORF abundance in ribosome space for human glia (hsa glia n = 53,403) in ORFs per million (OPM) by ORQAS and P sites per nucleotide per million (P sites pNpM) by ORFquant, indicating whether each ORF is predicted only by ORQAS (orange) or by ORFquant (yellow), by both (blue), or none of them (gray). Other samples are shown in Supplementary Fig 8. **h** Distribution of the average RNA expression in human glia (has glia n = 30,035) in TPM for ORFs predicted only by ORQAS (orange), by ORFquant (yellow) or by both (blue). Other samples are shown in Supplementary Fig. 9.

isoforms corresponded to either single-isoform genes or to the isoform with the highest expression in a sample (main isoform) (Fig. 1d). However, from those genes with multiple isoforms expressed at the RNA level, 3,471–3,570 (52.6–55.5%) of genes in human and 577–898 (27.6–34%) in mouse had an alternative isoform translated (Fig. 1d). From all translated isoforms, 47.3–49.2% in human, and 28.3–34.9% in mouse, correspond to alternative isoforms (secondary or other isoform, Fig. 1d). In genes with multiple isoforms, the main isoform showed the highest average Ribo-seq coverage compared with secondary isoforms, albeit not as high as for the single-ORF genes used as positive controls (Fig. 1e). In addition, the periodicity of the translated isoforms was uniform across the ORFs (Supplementary Fig. 5).

We compared ORQAS with ORFquant[38], another method to calculate translation at isoform level from Ribo-seq. We run ORFquant on the same samples analyzed with ORQAS. Looking at the annotated ORFs, ORQAS detected more genes with multiple ORFs consistently across all samples tested (Fig. 1f, Supplementary Fig. 6). Calculating the level of agreement between both methods with a Jaccard index, there was a higher level of agreement at gene level than at isoform level (Supplementary Fig. 7). We also compared the translation abundance provided by both methods. ORFquant abundance is based on the normalized number of P sites per nucleotide, whereas ORQAS provides an abundance in ORFs per million (OPM), akin to TPM units. We observed in general a good correlation of the abundance values for ORFs predicted to be translated by both methods (Fig. 1g, Supplementary Fig. 8). However, many isoforms that did not pass ORQAS read-count filter still showed high abundance according to ORFquant. ORFs predicted only by ORFquant turned out to have low or no RNA-seq expression (Fig. 1h, Supplementary Fig. 9).

**ORQAS discriminates translation at isoform level**. To validate the translation predictions by ORQAS at isoform level, we compared the results from Ribo-seq with the RNA abundances measured from polysomal fractions in the same human neuronal and embryonic stem cell samples[39]. ORQAS predicted 27,552 translated isoforms in stem cells, and 25,034 in neurons (Supplementary Fig. 10a). To control for the fact that monosomes can contain translating short mRNAs[40] (Supplementary Fig. 10b, c), we separated isoforms in three different length ranges. We found that translated isoforms predicted by ORQAS were enriched in polysomal fractions at all length ranges (Fig. 2a, Supplementary Fig. 10d). In contrast, isoforms with RNA expression, but not predicted to be translated with ORQAS, were enriched in monosomal fractions (Fig. 2a). This provides support for our predictions and is also consistent with a small proportion of our predicted translated isoforms to be associated with nonsense-mediated decay (NMD) targets, which are generally associated with monosomes[41].

Cross-species conservation is a strong indicator of stable protein production[42]. We thus decided to test the conservation of our translated isoforms in human and mouse, using glia and glioma samples available for both species. To this end, we used an optimization method to determine the human–mouse protein isoform pairs most likely to be functional orthologs (Fig. 2b, see the "Methods" section). From 15,824 human–mouse 1-to-1 gene orthologs, we identified 18,574 human–mouse protein isoform pairs, and 7112 (64%) of the 1-to-1 gene orthologs had more than one such isoform pair. These 18,574 protein pairs represent orthologous protein isoform pairs. We found that orthologous isoform pairs were significantly enriched in translated isoforms in both species (Fisher's test p-value <2.2e−16 in all datasets) (Fig. 2c), providing further support for our predictions.

To perform an additional validation of our findings, we considered isoform-specific regions (Fig. 3a), since evidence mapped to these regions can then be unequivocally assigned to the isoform (see Supplementary Figs. 11 and 12 for specific examples). We defined two types of isoform-specific regions. One type was defined in terms of isoform-specific nucleotide sequences, i.e., continuous nucleotide stretches that are only included in an isoform. From the annotation, we were able to identify 34,554 isoforms with such regions in human and 29,447 in mouse. We found that translated isoforms had a significantly higher density of Ribo-seq reads per nucleotide in those regions than non-translated isoforms (Fig. 3b, Supplementary Fig. 13a). In addition, unique sequence regions harbored more uniquely mapping Ribo-seq reads in translated isoforms compared with non-translated ones (Fig. 3c, Supplementary Fig. 13b). Overall, we were able to validate 56–80% of the isoform-specific sequence regions.

To be able to validate our predictions using P sites and peptides from MS experiments, we additionally considered isoform-specific ORF regions (Fig. 3a). These were defined as sequences that may or may not be shared between isoforms, but had a specific frame in each isoform, so that peptides from MS experiments can be unequivocally mapped on these regions. From the annotation, we found in total 44,299 isoforms with specific ORF regions in human and 34,329 in mouse. These included the 34,554 and 29,447 isoforms calculated before with ORFs from isoform-specific sequences in human and mouse, respectively: hence, there were 9745 and 4882 isoforms in the human and mouse annotations, respectively, that did not have differences in sequence, but had different overlapping ORFs. We found that translated isoforms had a significantly higher density of P sites in isoform-specific ORFs (Fig. 3d, Supplementary Fig. 13c). We further used peptides from MS experiments[42] to validate our predictions. Overall, we validated more isoforms predicted as translated compared with those predicted as not-translated (Fig. 3e). The rate of validation decreased with the region length, as expected for MS experiments[41]. On the other hand, the proportion of predictions validated with peptides increased using increasing cutoffs for RNA expression (Supplementary Fig. 13d). Overall, we were able to validate 48–73% of the isoform-specific ORFs tested.

In summary, from all the protein-coding transcript isoforms considered from the annotation (84,024 in human and 48,928 in mouse), 58–59% in human and 63–65% in mouse showed RNA expression >0.1 TPM (Supplementary Table 4, translated isoforms are available in Supplementary Data 1). From these expressed isoforms, about 40% in human and 41–54% in mouse were predicted to be translated by ORQAS, and 23–43% were validated using independent data, including conservation (Fig. 3f). Furthermore, 60% of the alternative isoforms predicted as translated had independent evidence of translation, and these corresponded to approximately 10% of all the annotated alternative isoforms in human and mouse (Supplementary Fig. 13e). Our analyses thus indicate that alternative transcript isoforms are often translated into protein, although they represent a small fraction of all expressed transcripts.

**Conserved impact of differential splicing on translation**. Differential splicing is often assumed to lead to a measurable difference in protein production. However, this has only been shown for a limited number of cases[16]. We addressed this question at genome scale using our more sensitive approach based on Ribo-seq. We used SUPPA[43,44] to obtain 37,676 alternative splicing events in human and 17,339 in mouse that covered protein-coding regions (see the "Methods" section, all alternative splicing

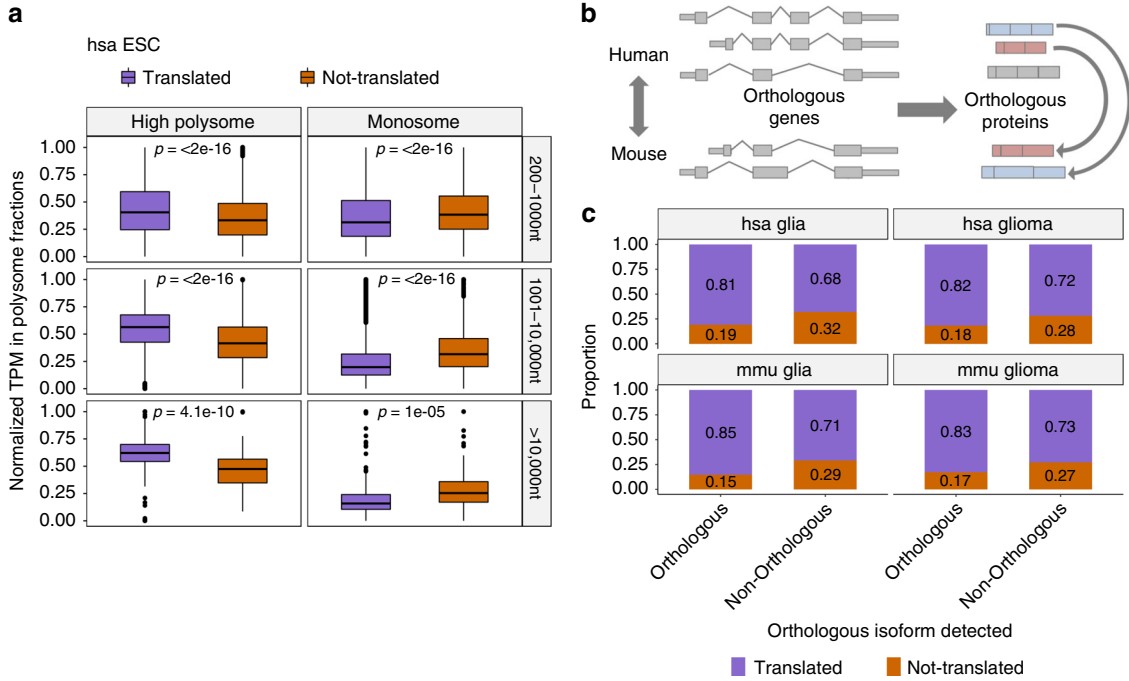

**Fig. 2 Validation of predictions. a** We show the distribution of the relative abundance in high polysome (left panels) and monosome (right panels) fractions of human embryonic stem cells (ESC) for translated isoforms and for isoforms with RNA expression (TPM > 0.1) but predicted as not-translated. The plot shows the results for three different ORF lengths: 200–1000nt (high polysome Wilcoxon test p-value 7.2e−67, translated $n = 18{,}056$ and not-translated n=9397, and monosome Wilcoxon test p-value 2.3e−77, translated $n = 15{,}023$ and not-translated $n = 8277$), 1001–10000nt (high polysome Wilcoxon test p-value 2.5e−242, translated $n = 19{,}169$ and not-translated $n = 3813$, and monosome Wilcoxon test p-value 1.1e−94, translated $n = 16{,}002$ and not-translated $n = 3179$), and longer than 10000nt (high polysome Wilcoxon test p-value 5.1e−10, translated $n = 200$ and not-translated $n = 69$, and monosome Wilcoxon test p-vaue 5e−06, translated $n = 178$ and not-translated $n = 66$). The results for neuronal cells are given in Supplementary Fig. 3. Box boundaries correspond to the first and the third quartiles, the median is indicated by a thick black line, top and bottom whiskers extend up to 1.5 times the interquartile range to the highest and smallest values, respectively, and outliers are indicated as black dots. **b** Cross-species conservation of protein isoforms. Protein isoforms from each 1-to-1 orthologous gene pair are compared, and candidate orthologous pairs are assigned using an optimization approach (see the "Methods" section). **c** For the set of ORFs encoding a human–mouse orthologous protein pair (orthologous) and for those encoding proteins without an orthologous pair in mouse (non-orthologous), we plot the percentages that are predicted to be translated (translated) and the ones with RNA expression (TPM > 0.1) but predicted as not-translated (not-translated). We show here the results for human glia (Fisher's test p-value = 1.41e−140, orthologous $n = 12{,}801$ and non-orthologous $n = 17{,}111$) and glioma (Fisher's test sp-value = 3.63e−85, orthologous $n = 12{,}453$ and non-orthologous $n = 16{,}278$), and for mouse glia (Fisher's test p-value = 1.143e−130, orthologous $n = 12{,}792$ and non-orthologous $n = 7356$) and glioma (Fisher's test p-value = 7.462e−53, orthologous $n = 12{,}015$ and non-orthologous $n = 5905$). Other mouse samples are shown in Supplementary Fig. 10e.

events are available in Supplementary Data 2-4). Using the same SUPPA engine to convert isoform abundances to event inclusion values[43,44], we calculated a relative abundance (RA), defined as the proportion of translation abundance of the isoforms given by ORQAS that is explained by a particular alternative splicing event (Fig. 4a). Accordingly, in analogy to a relative inclusion change (ΔPSI) in RNA space, we were able to measure the relative differences in ribosome space due to the inclusion or exclusion of particular alternative exons, or ΔRA.

Comparing the glia and glioma samples in human, we found 856 events with a significant change in RNA splicing ($|ΔPSI| > 0.1$ and p-value < 0.05, as calculated by SUPPA), and 590 events with significant differential translation ($|ΔRA| > 0.1$ and p-value < 0.05, as calculated by SUPPA), with a significant overlap of 363 events between them (Jaccard index z score = 89.386 compared with the Jaccard index distribution of the overlaps from 1000 subsample sets of the same size) (Supplementary Fig. 14a). Similarly, in mouse, we found an overlap of 179 events (Jaccard index z score = 65.326) between 471 events with a significant change in RNA splicing ($|ΔPSI| > 0.1$ and p-value < 0.05, as calculated by SUPPA) and 240 with significant change in translation ($|ΔRA| > 0.1$ and p-value < 0.05, as calculated by SUPPA) (Supplementary Fig. 14b).

We observed a concordance in the direction and magnitude of the change in significant events in RNA and ribosome space in human (Pearson R = 0.9904, p-value = 5.309e−312) and mouse (Pearson R = 0.9937, p-value = 2.113e−170), and this concordance was greater than for the rest of events (Fig. 4b, Supplementary Fig. 14c). We further observed a similar proportion of event types changing significantly in RNA and ribosome space, with an enrichment of exon-skipping events in human (Fig. 4c) and mouse (Supplementary Fig. 14d). To investigate the nature of these enriched cases, we considered very short alternative exons, or microexons, which are known to be differentially included in brain cells[23,45]. We tested the inclusion properties of alternative exons in RNA and ribosome space in our samples. We observed that alternative exons show less inclusion in glioma compared with glia at lengths 51nt or below, with a stronger pattern below 28nt, which are the two previously employed length cutoffs to define microexons (Supplementary Fig. 14e)[23,45].

Microexons, defined here as exons of length 51nt or less, were enriched in the events with significant changes in RNA-seq (Fisher's test p-value 0.01125200 in human, 1.231482e−10 in mouse) and in Ribo-seq (Fisher's test p-values 0.03917744 in human, 1.049473e−06 in mouse). This enrichment was also

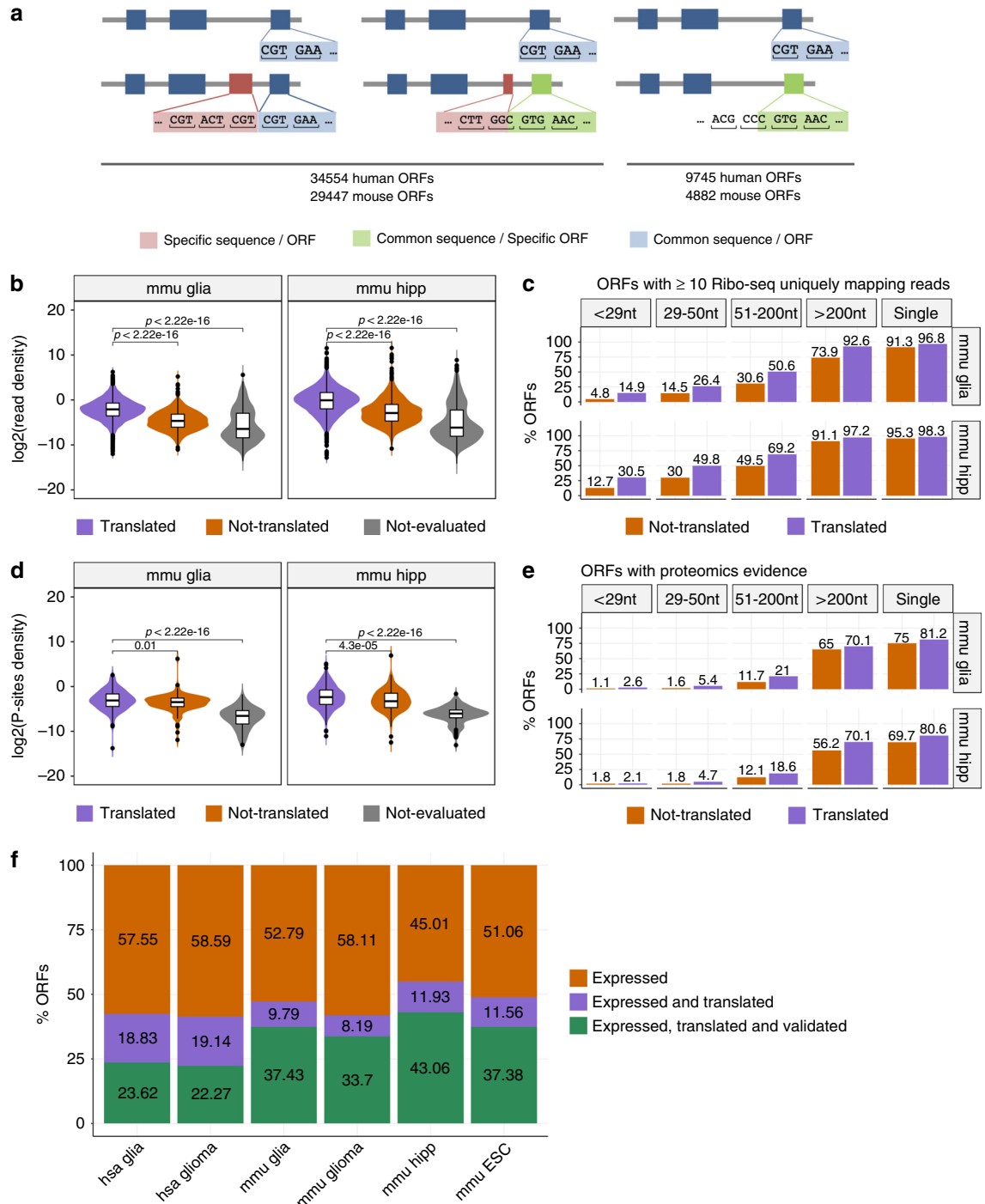

present if microexons were defined to be of length <28nt: Glia vs Glioma in human (Fisher's test p-value for RNA-seq 5.435e–13, for Ribo-seq 1.17e–09), Glia vs Glioma in mouse (Fisher's test p-value for RNA-seq 7.47e–14, for Ribo-seq 3.194e–06), and Human ESC vs differentiated neurons (Fisher's test p-value for RNA-seq 2.725e–08, for Ribo-seq 6.768e–06). Moreover, microexons were enriched in events decreasing inclusion in glioma compared with glia in both human (Fisher's test p-values 1.382e–12 for RNA-seq and 5.602e–10 for Ribo-seq) (Fig. 4d) and mouse (Fisher's test p-values 6.386e–16 for RNA-seq and 3.446e–06 for Ribo-seq) (Supplementary Fig. 14f). We repeated the same analysis using data from human neuronal differentiation[39] and found that microexons were also enriched in the comparison between embryonic stem cells and neuronal cells in

terms of RNA splicing and translation (Fisher's test p-values 8.435e–06 for RNA-seq and 6.597e–05 for Ribo-seq) (Supplementary Fig. 14g). Furthermore, using RNA sequencing from polysome fractions from the same stem cell and neuronal samples, we were able to validate the change in inclusion patterns of microexons under the same conditions (Fig. 4e).

We compared the capacity of ORQAS and ORFquant to detect translation of microexon-containing ORFs. In general, we observed that isoforms predicted by ORQAS included more cases with short (<51nt) and very short (<29nt) isoform-specific regions than ORFquant (Supplementary Fig. 15), hence providing a greater potential to identify translating microexons. Indeed, ORQAS identified more microexons than ORFquant in all tested samples (Fig. 4f). We also performed a comparison with

**Fig. 3 Validation with isoform-specific regions. a** Isoform-specific sequence regions (pink) are defined as the parts of an isoform ORF not present in any other isoform from the same gene. Isoform-specific ORFs (green) are defined as a region shared between two isoforms, but with a different frame in each isoform. **b** Density of Ribo-seq reads per nucleotide over the isoform-specific sequence regions for isoforms with ≥1 unique mapping read for mouse glia (mmu glia n = 13,541) and hippocampus (mmu hipp n = 18,015) samples. Distributions are given for predicted translated isoforms, for isoforms that did not pass the threshold of uniformity and periodicity (not-translated), and for the isoforms with low expression (TPM < 0.1) (not evaluated). Other samples are shown in Supplementary Fig. 13a. Box boundaries correspond to the first and the third quartiles, the median is indicated by a thick black line, top and bottom whiskers extend up to 1.5 times the interquartile range to the highest and smallest values, respectively, and outliers are indicated as black dots. **c** Percentage of regions with ≥10 uniquely mapping Ribo-seq reads in isoform-specific sequence regions, for mouse glia (mmu glia) and hippocampus (mmu hipp) samples. Other samples are shown in Supplementary Fig. 13b. **d** Density of Ribo-seq reads per nucleotide over the specific ORF regions for isoforms with ≥1 P-site position count, for mouse glia (mmu glia n = 1143) and hippocampus (mmu hipp n = 1445) samples. Other samples are shown in Supplementary Fig. 13c. **e** Percentage of sequence- and ORF-specific regions with one or more mass-spectrometry peptides, separated according to region length, for mouse glia (mmu glia) and hippocampus (mmu hipp) samples. Box boundaries correspond to the first and the third quartiles, the median is indicated by a thick black line, top and bottom whiskers extend up to 1.5 times the interquartile range to the highest and smallest values, respectively, and outliers are indicated as black dots. **f** Proportion of isoforms expressed (TPM > 0.1) predicted to be translated and validated with one or more sources of evidence: conservation, unique regions, and counts per base or peptides in specific ORF regions.

previously detected alternative splicing events using a direct mapping of Ribo-seq reads to the exon–exon junctions[31]. While there was a high overlap for exons longer than 51nt, ORQAS identified more microexons (Fig. 4g).

Our results provide evidence that differential splicing leads to a qualitative and quantitative change in the protein products from a gene locus. These results are also consistent with a functional relevance of the inclusion of microexons in protein-coding transcripts in neuronal differentiation and their inclusion loss in brain-related disorders[22,23]. Our analyses also highlight the strength of ORQAS in detecting these microexons compared with other alternative methods.

To further test the relevance of our findings, we considered a set of 1487 alternative exons conserved between human and mouse (Fig. 5a). A high proportion of them changed in the same direction between glia and glioma (66% in RNA-seq and 78% in Ribo-seq). Moreover, we observed that among the events with concordant changes between both species, there was an enrichment of microexons, with a significant trend toward less inclusion in glioma (Fisher's test p-values 5.389e−05 for RNA-seq and 5.521e−04 for Ribo-seq) (Fig. 5b). Furthermore, there was a correlation between the changes in ribosome space in human and mouse (Fig. 5c).

Among the microexons with a differential pattern of splicing and translation, we identified one in the gene GOPC (Fig. 5d), which was previously linked to glioblastoma[46], and one in the gene CERS6 (Fig. 5d), which has been associated with chemotherapy resistance[47]. To test further the potential relevance of the identified microexons with conserved differential pattern, we calculated their RNA splicing inclusion patterns across other normal and tumor brain samples. In particular, we analyzed samples from glioblastoma multiforme (GBM) from TCGA[48], neuroblastoma (NB) from TARGET[49] (Fig. 5e), and samples from the cortex and hippocampus from GTEX[50]. Microexons with a conserved impact on translation recapitulate the pattern of decreased inclusion in GBM compared with the postmortem normal brain regions (Fig. 5e). Differentially translated microexons may explain tissue differentiation as well as tumor-specific properties, as they separate out healthy tissues and tumor types (Supplementary Fig. 16a). In contrast, conserved microexons are clearly more prevalent in differentiated brain tissues than in tumor samples (Supplementary Fig. 16b).

## Discussion

We have described ORQAS (https://github.com/comprna/orqas), a method to obtain transcript abundance estimates at isoform level in ribosome space, to identify multiple protein products

from a gene, and to investigate differential translation associated with alternative splicing and differential transcript usage between conditions. Our approach presents several novelties and advantages over other methods[31,34,38,39]: (i) filtering based on the periodicity and uniformity of the Ribo-seq reads improves the identification of translated isoforms, (ii) the use of RNA expression to guide the translation abundance estimation avoids many potential false positives, (iii) our description in terms of isoforms allows the identification of more translated alternative isoforms and, in particular, microexons compared with other methods, (iv) our validation using isoform-specific regions, regardless of whether these regions could be encoded into a standard alternative splicing event, provides a more general validation of the impact of differential splicing on translation, and (v) ORQAS provides isoform abundance estimates in ribosome space that can be reused by other tools, like SUPPA. Our analyses were limited to transcripts and ORFs present in the annotation, but it is possible to adapt our method to work with new isoforms once their sequences and ORFs are predicted from the RNA-sequencing data. We also did not consider the contribution from upstream ORFs (uORFs), which can also be potentially detected based on abundance and periodicity[51].

We estimated that in total about 40–50% of the protein-coding isoforms with RNA expression showed some evidence of translation, and that around 20,700 proteins are produced in human and 13,000–17,500 in mouse in the tested conditions. In addition, about 5700–5800 genes in human and 2600–3900 in mouse produce more than one protein in those conditions. These estimates are considerably lower than what is generally predicted from RNA expression[8]. This may be explained by the limited coverage of Ribo-seq reads, but may also be due to the fact that RNA-seq artificially amplifies fragments of unproductive RNAs, leading to many false positives. Nonetheless, our data indicate that many more ORFs are translated in a given sample than what is detectable by current proteomics methods, and that the number of detected translated ORFs is close to estimates obtained using a combination of proteomics and sequence conservation[42]. Importantly, we found that multiple ORFs are translated from the same gene and at different abundances across conditions.

Around 40% of the events detected with differential RNA splicing showed consistent measurable changes in Ribo-seq in the same direction, which supports the notion that changes in RNA processing have a widespread impact on the translation of ORFs from a gene. In particular, we found that a pattern of decreased inclusion of microexons in glioma with respect to normal brain samples is recapitulated at the translation level, providing in vivo evidence that the splicing changes in microexons have an impact on protein production. Microexon inclusion is a hallmark of

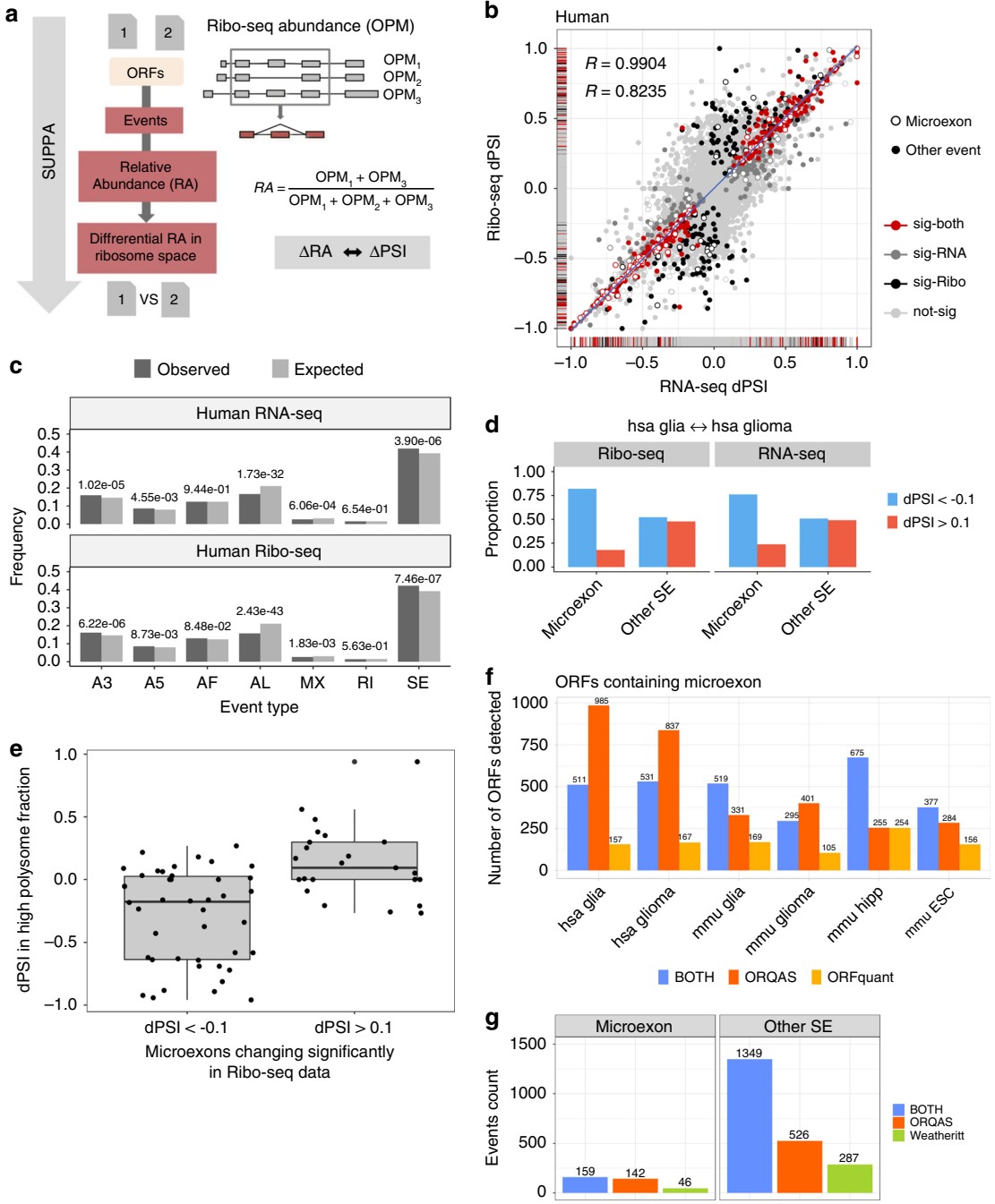

**Fig. 4 Differential translation linked to differential splicing. a** ORF abundances were used to calculate an event relative abundance (RA) with SUPPA. **b** Correlation of splicing and translation changes in events. In red, significant cases in ribosome and RNA space (363); in black (493) or dark gray (227), cases significant only in one comparison; in light gray, nonsignificant cases. The bands show the density of cases along each axis. Empty circles indicate microexons. In blue, the correlation of the red points. In gray, the correlation of all other exons (black/dark gray/gray), both including microexons. **c** Differentially spliced events in RNA or ribosome space by event type: alternative 3′ss/5′ss (A3/A5), alternative first/last exon (AF/AL), mutually exclusive (MX) exon, retained intron (RI), and skipping exon (SE). In light gray, the proportion of SUPPA events calculated from annotated coding regions in human. In dark gray, the proportion of events with significant change between glia and glioma. There is enrichment of SE events in RNA (Fisher's p-value 3.90e−06) and Ribo-seq (Fisher's p-value 7.46e−07), of A3 events in RNA (Fisher's p-value = 1.02e−05) and Ribo-seq (Fisher's p-value 6.22e−06), of A5 events in RNA (Fisher's p-value 4.55e−03) and Ribo-seq (Fisher's p-value 8.73e−03), and significant depletion of AL events in RNA (Fisher's p-value 1.73e−32) and Ribo-seq (Fisher's p-value 2.43e−43) and of MX events in RNA (Fisher's p-value 6.06e−04) and Ribo-seq (Fisher's p-value 1.83e−03). **d** Enrichment of microexons with an impact on RNA splicing and ORF translation in glia vs glioma. dPSI denotes the difference in relative abundance in RNA or ribosome spaces. **e** dPSI values for high polysome fraction between neuronal samples and ESCs (y-axis) for microexons with significant change in ribosome space (x-axis). Box boundaries correspond to the first and third quartiles, the thick black line indicates the median, and top and bottom lines indicate the maximum and minimum values. Black dots represent all data points. **f** ORFs containing microexons detected only by ORQAS (orange), ORFquant (yellow), or both (blue). **g** Microexons (<52nt) and other SE events detected in ribosome space with PSI > 0.1 only by ORQAS (orange), by analyses from ref. [31] (green), or by both methods (blue).

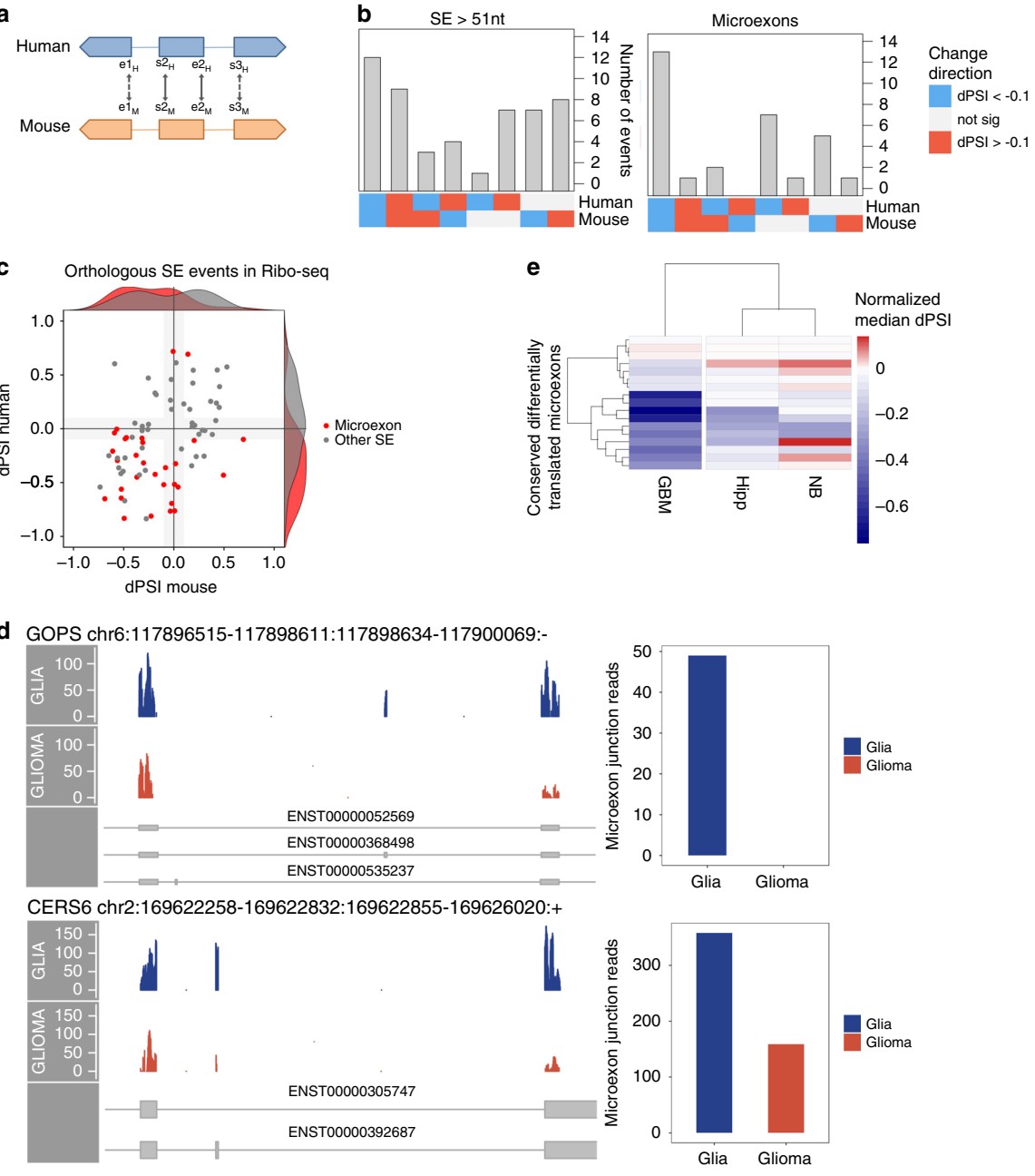

**Fig. 5 A conserved program of differential RNA splicing and translation. a** Conserved alternative splicing events were obtained by mapping with LiftOver the coordinates of the alternative exons (s2 and e2), and the internal coordinates of the flanking exons (e1 and s3). We considered those alternative exons that had at least (s2 and e2) conserved. **b** Directionality of the changes in conserved alternative exons longer than 51nt (left panel) and microexons (right panel) in ribosome space. As before, dPSI indicates here the difference in relative abundance in both RNA and ribosome space. The plot shows the number of events (*y*-axis) according to whether they were significant in human, mouse, or both, and the direction of change (*x*-axis). We indicate in blue if the event had a significant decrease in inclusion smaller than −0.1, and in red if the event had a significant increase in inclusion larger than 0.1. **c** Correlation between the difference in relative abundance (dPSI) in human on the *y*-axis and mouse on the *x*-axis for the conserved differentially spliced events in ribosome space. Microexons are depicted in red. **d** Examples of microexons that change significantly in RNA and ribosome space between glia and glioma. For the genes *GOPC* and *CERS6*, we show Ribo-seq reads mapping to the microexon region and its flanking exons (left panels) and the number of Ribo-seq reads crossing the microexon junctions (right panels) in both glia (in blue) and glioma (in orange). **e** Patterns of inclusion of conserved differentially translated microexons in normal hippocampus (Hipp) samples from GTEX, glioblastoma multiforme (GBM) from TCGA, and neuroblastoma (NB) from TARGET. The heatmap shows the difference of the median PSI with respect to the normal brain cortex tissue from GTEX.

neuronal differentiation[22,23,44], and glia partly recapitulates the pattern of microexon inclusion found in neurons[23]. The decreased inclusion of microexons observed in glioma suggests a dedifferentiation pattern similar to the one previously described before for other tumors[52], but could also be indicative of a difference in the content of neuronal cells in the samples compared.

In either case, the evolutionary conservation of the change at RNA expression and protein production indicates a conserved functional program in these samples.

Our capacity to predict RNA splicing variations from RNA-seq data currently exceeds our power to evaluate the significance of those events regarding protein production with current

proteomics technologies[53]. Despite this limitation, mass spectrometry can show for a small number of cases that splicing changes impact the abundance of proteins produced by a gene[16]. Our findings are in agreement with these results, and moreover overcome current limitations to determine genome-wide impacts of RNA-processing changes on protein production. Furthermore, our analyses indicate that the majority of translated alternative isoforms shows less than 25% variation in length with respect to the most highly expressed isoform, suggesting that for most part, the functional impacts from alternative splicing are mediated by slight modifications in the protein sequences[25], rather than through the production of essentially different proteins. In summary, ORQAS leverages ribosome profiling to provide a genome-wide coverage of genes and transcript isoforms, and allow a more effective testing of the impacts of splicing on protein production, as well as the identification and validation of multiple proteins from the same gene locus.

## Methods

**Preprocessing of RNA- and Ribo-seq datasets**. RNA- and Ribo-seq datasets were downloaded from Gene Expression Omnibus (GEO) for the following samples: normal glia and glioma from human and mouse (GSE51424)[30], mouse hippocampus (GSE72064)[35], mouse embryonic stem cells (GSE89011)[36], and three steps of forebrain neuronal differentiation in human (GSE100007)[39]. Adapters in RNA- and Ribo-seq datasets were trimmed using cutadapt v.1.12 with additional quality filters for RNA-seq (−q = 30). We further discarded reads that mapped to annotated rRNAs and tRNAs. The remaining reads in RNA- and Ribo-seq datasets were filtered by length (≥26 nucleotides).

**Quantification of transcripts and open-reading frames**. We used the Ensembl annotation v85 for human (hg19) (GRCh37.85) and mouse (mm10) (GRCm38.85) by removing pseudogenes, short isoforms (<200 nt), and annotated isoforms with incomplete 5′ or 3′ regions. For the analysis of RNA-seq data, we used Salmon v0.7.2[54] to quantify transcript abundances in transcripts per million (TPM) units using the annotation of unique open-reading frames (ORFs). To quantify coding sequences (CDS) at the isoform level with the Ribo-seq data, we applied a modified version of Ribomap[34] that uses the RNA-seq abundance as priors for the optimization algorithm to distribute Ribo-seq reads among the different isoforms. As default, Ribomap uses the RNA-seq reads aligned to the transcriptome sequences with STAR[55]. Instead, we provided a direct quantification of the ORFs with RNA-seq using Salmon, to be used as priors by RiboMap. We calculated the translation abundances of each ORF based on Ribo-seq reads in ORFs per million (OPM) units, similar to TPM units, but for ORFs instead of complete transcripts and using Ribo-seq reads. OPM values are calculated as follows:

$$\text{OPM}_i = 10^6 \frac{n_i/l_i}{\sum_j n_j/l_j} \quad (1)$$

where $n_i$ is the estimated Ribo-seq counts in ORF $i$ and $l_i$ is the effective length of the same ORF.

We performed benchmarking against ORFquant[38] with default parameters, considering only the ORFs in the annotation with the above-mentioned filters.

**Identification of translated isoform-coding sequences**. We identified actively translated ORFs by calculating two parameters' read periodicity and read uniformity[33]. The periodicity is based on the distribution of the reads in the annotated frame and the two alternative ones. In order to calculate the read periodicity, we previously computed the position of the P site, corresponding to the tRNA-binding site in the ribosome complex. This was obtained by calculating the distance between annotated ATG start codons and the leftmost position covered by Ribo-Seq reads, for each read length. The uniformity was measured as the proportion of maximum entropy (PME) defined by the distribution of reads along the ORF

$$H(X) = \sum_{i=1}^{n} \left(\frac{N_i}{N}\right) * \log_2\left(\frac{N_i}{N}\right) \quad (2)$$

$$\text{PME} = \frac{H(X)}{\max(H)} \quad (3)$$

where $N$ represents the total number of reads, $N_i$ is the number of reads in each region $i$, and $\max(H)$ is the entropy assuming that the reads are equally distributed across the ORF. The maximum value is 1 and indicates a completely even distribution of reads across codons. These values were obtained for each sample by pooling the replicates and only considering ORFs with ten or more assigned Ribo-seq reads, and with RNA-seq abundance TPM > 0.1.

**Polysomal fraction analysis**. We used RNA-seq from high polysomal, low polysomal, and monosomal fractions from embryonic stem cells and neuronal cell culture in human (GSE100007)[39] to quantify isoforms with Salmon[54]. Only ORFs from protein-coding isoforms were used for quantification. For each isoform, we calculated the polysomal relative abundance as before[17] by dividing the abundance in each polysomal fraction in TPM units, by the sum of abundances in (high and low) polysomes and monosomes.

**Validation of isoform-specific regions**. We defined two different types of isoform-specific regions that were analyzed differently. Isoform-specific sequences are regions with a unique nucleotide sequence among the isoforms of the same gene. Isoform-specific ORFs are defined as regions that will give rise to different amino-acid sequences within the proteins of the same gene, either because of the presence of isoform-specific sequences or frame-shifted common sequences (Fig. 3a). According to the annotation, we identified 34553 isoforms containing isoform-specific sequences in human and 29,447 in mouse, and 44,298 isoforms containing isoform-specific ORFs in human and 34,329 in mouse. For the validation of isoform-specific sequences, we considered uniquely mapping Ribo-seq reads from the STAR output falling entirely inside these regions or in the junction of the specific sequence with the common region. Read densities inside those regions were calculated as the total number of uniquely mapping reads in the region divided by the length of the isoform-specific sequence. The validation of isoform-specific ORFs instead was performed using the profiles of counts in each base of the ORF, considering the expected position of the P site. For isoform-specific ORFs, the read densities were established as the total number of counts in the region divided by the length in nucleotides of the isoform-specific ORFs.

**Proteomics evidence in translated isoform-coding sequences**. We mined the proteomics database PRIDE[56] to search for peptide matches to ORFs. We only considered peptide datasets from mouse corresponding to tissues analyzed in this study: brain (PRD000010, PXD000349, and PXD001786), hippocampus (PRD000363, PXD000311, and PXD001135), and embryonic cell lines (PRD000522). This corresponded to a total of 328,200 peptides. We searched for peptide matches in translated ORFs and only kept peptides that had one perfect match to an ORF and did not have a match with 0, 1, or 2 amino- acid mismatches to any other annotated ORF isoform from the same or different genes.

**Differential inclusion of events in RNA and ribosome space**. We used SUPPA[43,44] to generate alternative splicing events defined from protein-coding transcripts and covering the annotated ORFs with *generateEvents* subcommand. The relative inclusion of an event was calculated as a Percent Spliced In (PSI) value with SUPPA in terms of the transcript abundances (in TPM units) calculated from RNA-seq and in terms of the ORF abundances (in OPM units) calculated from Ribo-seq with *psiPerEvent* subcommand (Fig. 4a). The test for significant differential inclusion of the events was applied in the same way for both cases by testing the difference between the observed change between conditions and the observed change between replicates using *diffSplice* subcommand[44].

**Calculation of orthologous isoforms**. We considered the set of 1-to-1 orthologous genes between human and mouse from Ensembl (v85)[57]. For each pair of orthologous genes, we calculated all possible pairwise global alignments between the human and mouse proteins encoded by these genes using MUSCLE[58]. For each alignment, we defined a score as the fraction of amino-acid matches over the total length of the global alignment and kept only protein pairs with score > = 0.8. From all the remaining protein pairs in each orthologous gene pair, we selected the best human–mouse protein pairs using a symmetric version of the stable marriage algorithm[59].

**Comparison with normal and tumor tissues**. We downloaded the transcriptome (GRCh38) TPM values calculated with RSEM[60] from the XENA browser (https://xenabrowser.net/datapages/) for GTEX[50], TARGET[49], and TCGA[48]. PSI values for the alternative splicing events in human were calculated with SUPPA[43,44] from these TPM values, and the coordinates of the events were transformed to GRCh37 (hg19). We used 105 cortex and 84 hippocampus samples from GTEX, 171 glioblastoma multiforme samples from TCGA, and 162 neuroblastoma samples from TCGA to analyze the PSI values of the identified translated microexons.

**Reporting summary**. Further information on research design is available in the Nature Research Reporting Summary linked to this paper.

## Data availability

Predicted translated isoforms in the samples analyzed together with their validation using independent datasets are provided in Supplementary Data 1. All calculated alternative splicing events in RNA and ribosome space with annotations for microexons and orthology are provided in Supplementary Data 2 (comparison between human glia and glioma), Supplementary Data 3 (comparison between human ESCs and neuronal cells), and Supplementary Data 4 (comparison between mouse glia and glioma). All data are available from the corresponding author upon reasonable request.

## Code availability

ORQAS pipeline is available at https://github.com/comprna/ORQAS/.

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

## Acknowledgements

We are grateful to T. Preiss and N. Shirokikh for useful discussions, to R. Weatheritt for providing access to his datasets, and to A. Closa for help obtaining the event data from cancer and normal tissue samples. We acknowledge funding from the Spanish Government and FEDER with grants BFU2015-65235-P, BIO2017-85364-R, and MDM-2014-0370, and by Catalan Government (AGAUR) with grant SGR2017-1020. MR-S had funding from an FI grant from the Catalan Government with reference 2018FI_B1_00126 for part of this work.

## Author contributions

M.R-S. designed the pipeline and analyzed the data. J.R-O. assisted in data collection and preprocessing and proteomics analysis. E.E. and M.M.A. supervised the project and provided guidance. The paper was written by M.R-S and E.E. with essential inputs from all authors.

## Competing interests

The authors declare no competing interests.
