## [Peer Review File · Nature Communications]

Reviewers' comments:

Reviewer #1 (Remarks to the Author):

In this manuscript, the authors describe their new pipeline for open reading frame (ORF) quantification for alternative splicing, ORQAS. It is able to quantify the translation of individual transcript isoforms using Ribosome profiling data. They address the challenge that detection of protein variation derived from differential microexon using unbiased proteomics is nowadays not possible, whereas their pipeline is able to detect conserved microexons between human and mouse. The highlight of their pipeline is how they determine the potential of ORF translation, by calculating the uniformity and the periodicity along the ORF. This paper lacks comparison with other computational methods (e.g RiboMap and SaTAnn), testing of the uniformity and periodicity parameters, and additional validations such as quantification between cytoplasmic and nuclear extracts) to be considered as a novel method for alternative splicing quantification. Furthermore, the description of many of the figures makes interpretation of the results difficult to evaluate and the methods section requires more extensive description.

Algorithm outline

- 1) Quantifies the abundance of ORFs in from RNAseq (TPMs).
- 2) Assigns ribosome sequencing (Ribo-seq) to the same ORFs by RiboMap.
- 3) Calculates for each ORF two parameters for determining their potential translation: Uniformity (proportion of the maximum entropy of the read distribution) and the 3nt signal periodicity along the ORF.

Major concerns

- 1) The authors ORQAS algorithm relies primarily on a published pipeline from the Kingsford group (Wang et al. 2016, Bioinformatics). The innovation they've applied is to integrate in uniformity and periodicity cutoffs. However, they provide no evidence that these measures are an improvement over the approach used by Ribomap. Since the authors are releasing their own package, ORQAS, they should demonstrate the unique strength and advantage of their approach.
- 2) In addition, the authors need to show that their cutoffs for uniformity and periodicity improve their ability to reliably detect active translation and that they are not removing data unnecessarily.
- 3) Previous work both published (Weatheritt et al. 2016, Sterne-Weiler et al. 2014, Floor and Doudna. 2016) and in prepublication servers (Calviello et al. <https://www.biorxiv.org/content/10.1101/608794v1>) have identified alternative splicing variants engaged with polysomes or ribosomes. Furthermore, both Weatheritt et al. and Calviello et al. have analyzed ribosome profiling data and have provided estimates for the number of splice variants engaged by the ribosome. The authors explain the advantages of their approach but do not contrast their results, especially in light of controversy of splice variant identification in mass spec data.

4) We undertook a brief comparison to SaTAnn, the authors should expand on this. SaTAnn versus ORQAS. Both approaches used HEK293 data (<https://www.nature.com/articles/nmeth.3688> and GSM1306496)

ORFs with quantified translation	Number of genes	Number of genes with only one ORF translated
Number of genes with translation of multiple ORFs		
SaTAnn	~24,000	~15,000 9,138 (60.92%) >5,800
ORQAs	~20,709	Not explicitly mentioned 5,237- 4,766

(Fig 1c -1d) 52.3-54.9%
(Fig 1c and 1d)

5) In the section: Ribosome profiling discriminates translation abundance at isoform level. The authors' validate their approach by comparing translated isoforms predicted by ORQAs with RNAseq of polysomal fractions and found that their predictions are enriched in the polysomal fraction but not in monosomes (Fig 2c). This validation is not sufficient to support the translation prediction power of ORQA as active translation can also take place in monosomes (See Heyer and Moore, 2016). We propose they should compare between cytoplasmic and nuclear extracts.

6) The authors need to provide a more detailed methods section. For example, we advise that the authors elaborate further on how they use SUPPA to convert isoform abundances to event inclusion values together with rationales that validate such a conversion strategy.

7) The microexon section is interesting, especially the identification of microexons in glia samples. However, the majority of previous studies have used a cutoff of <30 nt for microexons (Irimia et al, 2014; Torres- Méndez et al, 2019). Does the enrichment still exist with this cutoff?

Minor concerns

- TPM value of 0.1 is very low. We suggest an alternative cutoff of at least 1 for robust interpretation.
- Fig 1b. Legend says: single ORF housekeeping genes are in blue but in the figure it suggests otherwise.
- In the section: Ribosome profiling discriminates translation abundance at isoform level
 - 1) The authors mention they have 15,824 human-mouse 1-1 gene orthologs, and identify 18,574 human-mouse protein isoform pairs representing functional orthologs. They then state to find 7,112 (64%) of the 1-1 gene orthologs had more than one orthologous isoform pair. We suggest the authors elaborate on how they reached this figure, as well as provide more information about the relevant plot.
 - 2) In the text, the p-value is as " $< 2.2 \times 10^{-16}$ ", whereas in the figure legend they show the exact p-values.
- Figures 3b and Supp. Fig 4a show the density of reads per nucleotide compared with other isoforms and suggest that "both region types in translated isoforms showed a higher density of reads per nucleotide compared with other isoforms"

○ We believe these figures do not necessarily agree with the aforementioned statement from the main text and require further elaboration in the Figure legends such as:

- Clarification of whether the specific sequence sets are independent.
- Present results of statistical test (eg. Wilcoxon rank sum test) for significant difference.

○ We also suggest the authors present the data with Ecdf plot to contrast the distributions between datasets.

● It is unclear how Figure 3e and 3f support statements from the main text. In particular, the statement of “63-65% in mouse with an RNA expression > 0.1 TPM” is difficult to relate to the data presented in Figure 3 without further elaboration. In addition, the authors should provide further information Figure 3f supporting the statement that “~ 10% of annotated AS in both human and mouse had evidence of translation and these represented 60% of all translated isoforms”, as unclear how these numbers were reached.

● Fig. 4a) in the section of RA calculation, the figure says “ $RA = OPM1 + OPM2$ ”. We believe that the authors' initial intention being to calculate $RA = OPM1 + OPM3$. If so, please correct this typo.

● Fig 4e and 4f are not sufficient to support a direct connection of microexons included in RNAseq to Ribo-seq. Inclusion of a plot supporting a strong correlation between RNAseq and Riboseq for microexons (as Figure 4c) is strongly advised.

● Fig 5b The authors mention: “high proportion of them changed in the same direction between glioma and glioma (66% in RNA-seq and 78% in Ribo-seq)”. The authors need to provide more information, as this result cannot be clearly inferred from Fig.

● Similar to the previous comment, the authors mention that microexons were enriched in both species with a general trend towards less inclusion in glioma, the figure (Fig. 5c) does not explicitly show that glioma has less inclusion of microexons. Clarification of the x-axis and how dPSIs are calculated in this specific plot are required to address this discrepancy.

Reviewer #2 (Remarks to the Author):

Summary

Reixachs-Sole et al have developed a new pipeline to better understand translation of mRNAs at the level of mRNA isoform rather than gene. They have gone to validate the results of using this in both human and mouse, with a variety of different data sets and approaches. By comparing differences in mRNA isoform abundances and translation they show that several micro-exon containing isoforms are regulated between glioma and glioma.

The aim to understand translation at the mRNA isoform level is admirable and represents an important step forward in linking mRNA processing and translation. Overall, I think this is a good study but many

aspects could benefit from improved explanations and examples to illustrate, especially for a general interest journal such as Nature Communications. The focus ends up being on differential expression of microexons, between glia and glioma, rather than differences in isoform abundance and translation, which seems the logical requirement for this new pipeline the authors developed here. This work is novel and is certainly interesting the gene expression field.

Specific points;

a) Authors mention that exon boundaries are frequently bound by RNA-binding proteins. However, the majority of these exon boundaries are at exon-intron boundaries in the nuclear. There is limited evidence to suggest that in spliced transcripts that make it out to the cytoplasm are more bound by RBPs than other parts of spliced transcripts.

b) The manuscript would benefit greatly from a more detailed explanation of the novel ORQAS method that the authors have developed. For example, it seems like the designation of Ribo-seq reads to isoforms is based on relative abundance of mRNA isoforms from RNA-Seq-is this true? Then these transcripts would be filtered based on whether corresponding ORFs make it through cut-offs for uniformity of ribosome profiling reads across the ORF and periodicity. My concern is that transcripts may fail to pass these two thresholds for reasons other than alternative ORF translation. Similar metrics have been used previously to define translation events, so it is not obviously why they pipeline presented here is able to deconvolute translation of ORFs from alternatively spliced isoforms.

c) Uniformity of periodicity could be used as a cut-off since changes on frame could indicate inconsistency caused but translation of ORF from alternative transcript.

d) In the section "Ribosome profiling discriminates translation abundance at isoform level" it is not clear what "combination of protein features" means in Fig 2b. It seems unsurprising that isoforms called translated have evidence of their protein expressed. It would be informative to generate a false discovery rate, for those isoforms that have no evidence of translation. It is not clear how mass spec, immunohistochemistry and uniprot data was treated to ensure that signal could be confidently assigned to a specific ORF isoform over another?

e) In the text, polysome association is described, whereas in Fig 2c, high polysomes are mentioned. What specific complexes were defined as high polysomes? How many transcripts were included in this monosome vs high polysome distribution? From RNA-Seq data it is not clear how we can be sure that this is transcript specific. If RT-qPCR with primers designed specifically for detection of specific isoforms match this same pattern?

f) The validation performed in Fig3a is excellent. But these sections would benefit greatly from examples illustrating these types of events.

g) There is very little explanation of many of the panels. For example, 3c): what was the aim of this over how mass spec data had be used to support ORF isoform translation?

h) It is not clear how S4 is different to Fig3? Is it same but just including human samples too?

i) Use of SUPPA in Fig4 to probe differential translation linked to differential splicing is really the most interesting part of the manuscript. Since one of the big questions in the field is whether certain spliced isoforms are preferentially translated. This analysis starts to address this. Fig 4c suggests this is generally not the case. The majority of analysis focuses on whether the translation of spliced microexons and their differential splicing. This is an important question and result. However, it is not clear whether this analysis was dependant on the original pipeline ORQAS, developed here.

j) The focus of results seemed to be on changes correlating between RNA-Seq and Ribo-Seq, especially in Fig 5, that are differential between different cell types. But given the manuscript aims to understand isoform translation it would be more appropriate to analyse more deeply events whose RNA-Seq and Ribo-Seq don't correlate. These are the situations, one would argue, that require understanding of isoforms at bot splicing and translation level.

k) In discussion "These estimates are far from" should be reworded to give indication of direction of change.

Reviewer #3 (Remarks to the Author):

The authors have developed a new method, ORQAS (ORF quantification pipeline for alternative splicing) to quantify isoform-specific translation abundance, and have applied their method to a number of different datasets including glia/glioma and ES cells/neurons.

One of the limitations of ribosome profiling is that it does not directly measure protein peptide abundance and assumes that engaged ribosomes are direct arbiters of protein levels. As the authors point out, this is not strictly true, and builds upon previous studies to attempt to examine open reading frames that are more likely to be translated. Their polysome data is convincing.

While I have some concerns about the novelty given that other studies examining alternative splicing and ribosome profiling have previously been published, I do think that the authors have significantly improved upon these studies and this study deserves to be published in Nature Communications.

I have a few queries:

1. In Figure 4C, the correlation between the ribo-seq and RNA-seq data is remarkably high, close to 1 - almost too good to be true. What happens to the datasets that are not significant, i.e.. what is the

correlation between non-significant alternative splicing events for which there is ribo-seq data and vice-versa? Presumably, the correlation would not be as significant and is important to show, as it demonstrates the power of their approach.

2. I would suggest that the authors improve their description of their method in the first results section, including on expanding on the descriptions of uniformity and periodicity and why the integration of which is a significant advance over ribosome profiling alone.

Reviewers' comments:

Reviewer #1 (Remarks to the Author):

In this manuscript, the authors describe their new pipeline for open reading frame (ORF) quantification for alternative splicing, ORQAS. It is able to quantify the translation of individual transcript isoforms using Ribosome profiling data. They address the challenge that detection of protein variation derived from differential microexon using unbiased proteomics is nowadays not possible, whereas their pipeline is able to detect conserved microexons between human and mouse. The highlight of their pipeline is how they determine the potential of ORF translation, by calculating the uniformity and the periodicity along the ORF. **This paper lacks comparison with other computational methods (e.g RiboMap and SaTAnn), testing of the uniformity and periodicity parameters, and additional validations such as quantification between cytoplasmic and nuclear extracts) to be considered as a novel method for alternative splicing quantification. Furthermore, the description of many of the figures makes interpretation of the results difficult to evaluate and the methods section requires more extensive description.**

Algorithm outline

- 1) Quantifies the abundance of ORFs in from RNAseq (TPMs).
- 2) Assigns ribosome sequencing (Ribo-seq) to the same ORFs by RiboMap.
- 3) Calculates for each ORF two parameters for determining their potential translation: Uniformity (proportion of the maximum entropy of the read distribution) and the 3nt signal periodicity along the ORF.

Major concerns

1) The authors ORQAS algorithm relies primarily on a published pipeline from the Kingsford group (Wang et al. 2016, Bioinformatics). The innovation they've applied is to integrate in uniformity and periodicity cut-offs. However, they provide no evidence that these measures are an improvement over the approach used by Ribomap. Since the authors are releasing their own package, ORQAS, they should demonstrate the unique strength and advantage of their approach.

To show that our approach provides an advantage over the approach used by Ribomap, we performed the following comparison. We selected genes with one single ORF annotated, so there is no ambiguous mappings of Ribo-seq reads. These are 1005 genes with one single ORF annotated, and non-overlapping with the single-ORF genes used as positive controls. For these genes we calculated the proportion of cases that have evidence of protein expression from immunohistochemistry (IHC) experiments from the human protein atlas (THPA):

The cases labelled “Not evaluated” are those that do not show enough RNA expression to be evaluated by ORQAS. The cases labelled “Not translated” are the predictions from Ribomap without any filters for periodicity or uniformity. In both cases there was an enrichment of cases without evidence of translation from THPA. In contrast, after imposing the thresholds (the cases labelled “Translated”), we observed an enrichment of cases with protein evidence at different levels. This is now shown in Figure 1c.

Additionally, to show the unique strength and advantage of our approach, we calculated the proportion of translated isoforms according to the minimum RNA expression cut-off, which is another one of our filters. We observed that the proportion of translated isoforms increases as a function of the expression cut-off:

We have added this plot as supplementary Figure.

To further show the robustness of ORQAS, we also generated plots with the proportion of translated isoforms validated by peptides as a function of the RNA isoform expression cut-off. They all increase with

higher expression cut-offs. Thus, using a higher expression cut-off translation can be validated with greater confidence:

We have also included this plot as supplementary Figure 13.

2) In addition, the authors need to show that their cutoffs for uniformity and periodicity improve their ability to reliably detect active translation and that they are not removing data unnecessarily.

As shown above, the cutoffs of uniformity and periodicity improve the ability to detect cases with evidence of translation. We have added these plots to the article and have included an explicit mention to this in the text (highlighted in blue).

3) Previous work both published (Weatheritt et al. 2016, Sterne-Weiler et al. 2014, Floor and Doudna. 2016) and in prepublication servers (Calviello et al. <https://www.biorxiv.org/content/10.1101/608794v1>) have identified alternative splicing variants engaged with polysomes or ribosomes. Furthermore, both Weatheritt et al. and Calviello et al. have analyzed ribosome profiling data and have provided estimates for the number of splice variants engaged by the ribosome. The authors explain the advantages of their approach but do not contrast their results, especially in light of controversy of splice variant identification in mass spec data.

In the original submission we cited (Weatheritt et al. 2016), (Sterne-Weiler et al. 2014), and (Floor and Doudna. 2016). The preprint Calviello et al. appeared after our preprint <https://www.biorxiv.org/content/10.1101/582031v2> and also after we submitted our work for peer review.

In the original submission we used data from (Blair et al. 2017) for human ESCs and differentiated neuronal cells to validate some of our findings for microexons. We also used the mapping of RNA sequencing reads from polysomal fractions, which is the approach of (Sterne-Weiler et al. 2014), as independent validation of our findings.

In the revised version we now include an exhaustive comparison with the data from Weatheritt et al. 2016 and with the method SaTAnn (see below).

In our manuscript we described the differences of our method with previous approaches (Weatheritt et al.): ORQAS estimates the engagement of ribosomes in transcript variants, measuring for each transcript

isoform independently their periodicity and uniformity. In contrast Weatheritt et al. mapped Riboseq reads directly to exon-exon junctions without considering whether their validity in terms of the transcript-isoform level expression, and Ribo-seq uniformity or periodicity. We argue that, since ribosomes scan transcript molecules, ORQAS provides a description that is easier to interpret in terms of translation of an mRNA isoform and also one that is closer to what is actually happening in the cell. SaTAnn considers directly the P-sites from Ribo-seq reads per exon, and combines exons into transcripts, without considering the RNA expression. In the revised manuscript we show that the translation quantification provided by ORQAS and SaTAnn is similar for isoforms predicted by both methods, but SaTAnn predicts translation also in isoforms that show no RNA expression (see below for more details of this comparison).

Another novelty of ORQAS with respect to previous studies is that we compare with mouse and find that the direction in which alternative splicing potentially impact translation is conserved in specific events, thereby establishing the potential functionality of these changes.

4) We undertook a brief comparison to SaTAnn, the authors should expand on this. SaTAnn versus ORQAS. Both approaches used HEK293 data (<https://www.nature.com/articles/nmeth.3688> and GSM1306496)

ORFs with quantified translation	Number of genes	Number of genes with only one ORF translated
Number of genes with translation of multiple ORFs		
SaTAnn	~24,000	~15,000 9,138 (60.92%) >5,800
ORQAs	~20,709	Not explicitly mentioned 5,237- 4,766

(Fig 1c -1d) 52.3-54.9%
(Fig 1c and 1d)

SaTAnn operates differently from ORQAS. SaTAnn works directly with P-sites, without taking into account RNA-seq abundances, it analyses first exons, which are then combined together. We have performed an exhaustive comparison with SaTAnn. To this end, we run SaTAnn with the same samples we analyzed in our manuscript. We have added these analysis to the manuscript.

We first compared the number of genes predicted to have at least 1 translated ORF (left panel below) and the total number of translated ORFs predicted (right panel below). We do not observed any systematic trend in these results. In some samples, ORQAS predict more translated genes and a more translated ORFs, and in other samples the trend is the opposite:

Interestingly, although both methods predict a similar number of genes in human glioma (hsa glioma), ORQAS predicts a larger number of ORFs. Also, although SaTAnn predicts more genes with translated ORFs in mouse glia (mmu glia), ORQAS predicts more ORFs overall. This suggests a difference in the number of translated ORFs per gene. To investigate this further, we calculated the number of genes with 1, 2, 3,... translated isoforms. We observed that ORQAS detects more genes with multiple ORFs:

This is now shown in Fig. 1f and in Supp. Fig. 6. We further calculated the agreement between both methods in terms of the genes translated and the total ORFs using a Jaccard Index = $\text{Intersection(ORQAS, SaTAnn)} / \text{Union(ORQAS, SaTAnn)}$. We observed a high level of agreement at gene level (60-80%) (left panel below), but slightly lower level of agreement at isoform-ORF level (right panel below) (now shown in Supp. Fig. 7):

We also compared the quantification provided by both methods. SaTAn provides a quantification based on the normalized number of P-sites per nucleotide (y axis below), whereas ORQAS provides a quantification in ORFs per million (OPM) (x axis below), akin to the TPM units. We observed that for the ORFs predicted as translated by both methods, the quantification values correlate ($R=0.8076$) (all plots now shown in Fig. 1g and Supp. Fig. 8):

However, there were a number of cases (in green in the figure above) that ORQAS did not predict because they had not sufficient reads or no reads at all, but are translated and with high abundance value

according to SaTAnn. Looking further at these differences, we observed that the ORFs predicted by SaTAnn that were not detected by ORQAS had low or no RNA-seq expression (plots now shown in Fig. 1h and Supp. Fig. 8):

Thus SaTAnn predicted ORFs as translated even if they had no evidence of RNA expression. Considering how short Ribo-seq reads are, it is possible that these cases originate from incorrect mapping and are thus false positives. Cases with too low or no RNA expression evidence are discarded by ORQAS, so these potential false positives are avoided.

We additionally tested the capacity of ORQAS and SaTAnn to detect short unique regions, i.e. regions that are specific to the isoform ORF. We observed that ORQAS is able to detect more ORFs with short unique regions (Now shown in Supp. Fig. 15):

The plot above shows the number of translated ORF predicted (y axis) as a function of the length of their unique regions (x axis). Thus ORQAS provides an advantage to predict translation in ORFs with short unique regions.

We also analysed the capacity of ORQAS and SaTAnn to detect microexon-containing isoforms, defining microexons as exons of length 51nt or shorter (more on this below). ORQAS recovered a larger number of microexon-containing isoforms across the different samples tested (now shown in Fig. 4f):

We also performed a comparison with the data from Weatheritt et al. 2016. To perform this comparison, we run ORQAS with the same Hek293 sample used in Weatheritt et al. 2016 and took all the events that were potentially measurable by both methods. From these events, we plot below those that were detected ($PSI > 0$ in Ribosome space) by both methods, or by each method independently (now shown in Fig. 4g):

We also calculated the expression in Ribosome space (OPM units) and in RNA space (TPM units) of the genes where SE events were detected. Weatheritt et al. also finds events for which we cannot find expression:

This highlights once again the potential artefacts that may appear by mapping Ribo-seq reads directly to the alternative splicing events, and the advantage of ORQAS approach to avoid these potential false positives.

5) In the section: Ribosome profiling discriminates translation abundance at isoform level. The authors' validate their approach by comparing translated isoforms predicted by ORQAs with RNAseq of polysomal fractions and found that their predictions are enriched in the polysomal fraction but not in monosomes (Fig 2c). This validation is not sufficient to support the translation prediction power of ORQA as active translation can also take place in monosomes (See Heyer and Moore, 2016). We propose they should compare between cytoplasmic and nuclear extracts.

Similar to translation on monosomes, there is also evidence of translation in the nucleus: <http://jcb.rupress.org/content/197/1/45>, so it is not entirely clear whether a comparison between genes expressed in the nucleus and cytoplasm will be necessarily conclusive.

Although translation can indeed occur in monosomes, we expect that mRNAs translating in monosomes should be generally short, as they would only hold 1 individual ribosome. First, we did not see any significant difference in the distribution of transcript lengths between monosome and high-polysome fractions in general (left plot below). We then separated between the cases that we defined as translated and not-translated. We observed that translating mRNAs are significantly shorter in monosomes (right plot below). This difference is not as large when considering not translated ORFs.

This comparison is now shown in Supp. Fig. 10. We thus separated previous Figure 2c according to three length ranges and observed that there was the same trends as before for translated transcripts in each of the length ranges (Now shown Fig. 2a):

As an additional test to show the robustness of ORQAS in defining translation at isoform level, we considered tissue specific genes as proxy for negative controls in a different tissue. We took genes annotated in THPA to have tissue specific expression in brain, heart, intestine, liver, spleen or testis, and calculated whether they were predicted as translated or not in our glia samples (now shown Supp. Fig. 3):

ORQAS on glia data predicts a higher proportion of translated genes in the subset of brain-specific genes compared with the subsets of genes specific in the other tissues.

6) The authors need to provide a more detailed methods section. For example, we advise that the authors elaborate further on how they use SUPPA to convert isoform abundances to event inclusion values together with rationales that validate such a conversion strategy.

We have tried to explain this better in the text. This conversion is supported by previous validations with RNA (see Alamancos et al. 2015, Trincado et al. 2018). With RNA, the abundance values from transcript isoforms summarized per event as a relative abundance (PSI) agrees with the measurements from RT-PCR using probes to capture the relative abundances for that event. For Ribo-seq we reasoned that, if Ribo-seq can be used to estimate the translation abundance of an isoform, we can also summarize these values per event and calculate a relative translation abundance. This represents the relative contribution of that particular exon to the translation abundance from a set of isoforms. That is, the relative contribution from an alternative exon to the translation of a gene.

7) The microexon section is interesting, especially the identification of microexons in glia samples. However, the majority of previous studies have used a cutoff of <30 nt for microexons (Irimia et al, 2014; Torres- Méndez et al, 2019). Does the enrichment still exist with this cutoff?

We confirmed our observations with exons of length <28nt by testing their enrichment in the set of alternative exons changing inclusion in RNA or Ribosome space between glia and glioma, and between neural and embryonic sample (now included in the text):

Comparison	Sequencing	Species	Fisher test p-value
------------	------------	---------	---------------------

Glia vs glioma	RNA-seq	human	5.435e-13
Glia vs glioma	Ribo-seq	human	1.17e-09
Glia vs glioma	RNA-seq	mouse	7.47e-14
Glia vs glioma	Ribo-seq	mouse	3.194e-06
hESC vs neural	RNA-seq	human	2.725e-08
hESC vs neural	Ribo-seq	human	6.768e-06

Irimia et al. 2014 justified the definition of microexon as <28nt from their observation that below this length there was an increased inclusion in neuronal samples with respect to non-neuronal samples. However, other length cut-offs have been used to define microexons (see e.g. Li et al. 2015, Ustianenko et al. 2017). To justify our choice of the <52nt cut-off, we looked at the distribution of inclusion levels (in PSI units) for exon-cassette events in RNA and Ribosome space, separated by exon-length ranges in glia and in glioma (now shown in Supp. Fig. 14e):

This plot shows that at length range [1,27] the differences of inclusion between glia and glioma is the largest, as expected for the definition of microexon from (Irimia et al. 2014). However, in the range (27,51] there is also a difference in inclusion between glia and glioma, which would support the inclusion of these exons into the definition of short exons with a brain cell specific inclusion pattern, or microexons.

Minor concerns

- TPM value of 0.1 is very low. We suggest an alternative cutoff of at least 1 for robust interpretation.

Using bins for different range of expression cut-off values, we observed that the higher the minimum expression considered, the larger the proportion of cases that are predicted as translated (now shown in Supp. Fig. 2):

However, the proportion of transcripts predicted as translated does not change much after removing cases below 1 TPM.

- Fig 1b. Legend says: single ORF housekeeping genes are in blue but in the figure it suggests otherwise.

We have fixed the figure legend.

- In the section: Ribosome profiling discriminates translation abundance at isoform level
 - 1) The authors mention they have 15,824 human-mouse 1-1 gene orthologs, and identify 18,574 human-mouse protein isoform pairs representing functional orthologs. They then state to find 7,112 (64%) of the 1-1 gene orthologs had more than one orthologous isoform pair. We suggest the authors elaborate on how they reached this figure, as well as provide more information about the relevant plot.

We have provided more information in the manuscript about this calculation and about the figures shown. Orthology is only annotated in databases at the gene level. However, we needed to obtain the pairs of potential ORF orthologs to establish conservation. We thus selected first the set of 15,824 1-1 gene orthologs between human and mouse, i.e. best reciprocal orthology assignments between human and mouse, hence no ambiguous mappings. We then needed to establish the orthology at protein isoform

level, i.e. the proteins in each pair of genes that can be considered protein orthologs. For that we considered all pairwise global alignments between human and mouse proteins, and selected the best possible pairs above a minimum score of 0.8 (defined as the fraction of amino acid matches over the total length of the global alignment), using a symmetric version of the stable marriage algorithm, as described before (Eyras et al. 2004). In this algorithm, pair assignments are established given an ordering of “preference” for each element, which is provided by the score calculated from the alignment. The algorithm produces optimal or “stable” pairs, in the sense that it provides the best available matches rather than the best reciprocal matches. That is, given a stable pair, there are no other pairs possible where both elements would prefer each other more than their current pair. This algorithm produces 18,574 human-mouse protein isoform pairs. We calculated then how these are distributed in genes orthologs: 36% of gene orthologs had one single protein isoform pair, and 64% had 2 or more protein isoform pairs.

2) In the text, the p-value is as “< 2.2 e-16”, whereas in the figure legend they show the exact p-values.

This was to avoid writing all p-values in the text and to indicate that all tests were significant and with p-values smaller than 2.2e-16. The text says “... were significantly enriched in translated isoforms in both species (p-value < 2.2e-16 in all datasets)”. The figure legend shows the actual p-values.

- Figures 3b and Supp. Fig 4a show the density of reads per nucleotide compared with other isoforms and suggest that “both region types in translated isoforms showed a higher density of reads per nucleotide compared with other isoforms”
 - We believe these figures do not necessarily agree with the aforementioned statement from the main text and require further elaboration in the Figure legends such as:
 - Clarification of whether the specific sequence sets are independent.
 - Present results of statistical test (eg. Wilcoxon rank sum test) for significant difference.
 - We also suggest the authors present the data with Ecdf plot to contrast the distributions between datasets.

We considered isoform-specific regions, since evidence mapped to these regions can then be unequivocally assigned to the isoform. We defined two types of isoform-specific regions. One type was defined in terms isoform-specific nucleotide sequences, i.e. continuous nucleotide stretches that are only included in an isoform. To validate our predictions with peptides from MS experiments and P-sites, we additionally considered isoform-specific ORF regions. These were defined as sequences that may or may not be shared between isoforms but had a specific frame in each isoform, so that peptides from MS experiments can be unequivocally mapped on these regions. These regions included the isoforms calculated before with the ORFs from the isoform-specific sequence.

Isoform-specific sequences are thus also isoform-specific ORFs, e.g. an alternative exon specific to an isoform defines an isoform-specific ORF region. Thus the two sets of isoforms are not independent. However, the datasets used in each case are independent. Isoform-specific ORFs were validated with P-sites and with mass-spec peptides, whereas isoform-specific sequences were validated with Ribo-seq reads, regardless of the position and identification of the P-site. We have tried to clarify this in the text, and added new plots. Significance is now shown in new plots in Figure 3 and in Supp. Fig. 13. Below we show the case for the densities of Ribo-seq reads in isoform-specific sequences:

We have modified the cartoon in Figure 3a to clarify the possible configurations of the isoform-specific regions. For improved clarity, we have also separated into independent panels the validation of isoform-specific sequences with Ribo-seq reads (now in Figs. 3b and 3c) and the validation of isoform-specific ORFs with P-sites or mass-spec peptides (now in Figs. 3d and 3e).

- It is unclear how Figure 3e and 3f support statements from the main text. In particular, the statement of “63-65% in mouse with an RNA expression > 0.1 TPM” is difficult to relate to the data presented in Figure 3 without further elaboration. In addition, the authors should provide further information Figure 3f supporting the statement that “~ 10% of annotated AS in both human and mouse had evidence of translation and these represented 60% of all translated isoforms”, as unclear how these numbers were reached.

Figure 3e is now shown in Fig. 3f, and Fig. 3f has been moved to Supp. Fig. 13e for clarity. Fig. 3f (previous Fig. 3e) shows the proportion of all expressed isoforms that are predicted to be translated and additionally have independent validation of translation. This validation is considered to be one or more of different sources of evidence: conservation, uniquely mapped Ribo-seq reads in isoform-specific sequences, and counts per base or peptides in isoform-specific ORFs. Supp. Fig. 13e (Previous Fig. 3f) shows the same information but splitting isoforms according to whether they are the main isoform or an alternative isoform, where “main” was defined as the isoform with the highest expression, and “alternative” are the rest of isoforms from the gene showing expression (TPM>0.1). The proportions shown were calculated by putting together all the results of the validation analyses and are available in Supp. Tables 3 and 4.

- Fig. 4a) in the section of RA calculation, the figure says “RA = OPM1 + OPM2”. We believe that the authors' initial intention being to calculate RA=OPM1 + OPM3. If so, please correct this typo.

That's right. We have corrected this error.

- Fig 4e and 4f are not sufficient to support a direct connection of microexons included in RNAseq to Ribo-seq. Inclusion of a plot supporting a strong correlation between RNAseq and Riboseq for microexons (as Figure 4c) is strongly advised.

We have included now in Figure 4b (and the Supplementary Figure for 14c mouse) also the cases that are not significant in RNA-seq or Ribo-seq or both in the comparison between glia and glioma, also highlighting (as empty circles) microexons:

The figure shows in red the cases that are significant in both Ribo-seq and RNA-seq space, in black or dark gray, the cases that are only significant in one case, and in light gray, the cases that are significant in neither of them. The density bands in the plot shows the distribution of cases along the axes. Empty circles indicate the microexons (red if significant, black/gray otherwise). In the inset in blue we give the correlation of the red points (including microexons), in gray we give the correlation of all other exons (black/dark-gray/gray, including microexons).

- Fig 5b The authors mention: "high proportion of them changed in the same direction between glioma and glioma (66% in RNA-seq and 78% in Ribo-seq)". The authors need to provide more information, as this result cannot be clearly inferred from Fig.

We have provided more information in the text to clarify this result. Additionally, for clarity we have put together Figures 5b and 5c (now Fig. 5b) and have improved the figure caption to make it more clear. In these barplots we show the events changing significantly in human and/or mouse in each direction: less inclusion (blue), more inclusion (red). We have also added a new plot (now Figure 5c) comparing the difference of inclusion values in human and mouse in Ribo-seq space, which shows that microexons have a conserved pattern of decreased inclusion in glioma. More details on this below.

- Similar to the previous comment, the authors mention that microexons were enriched in both species with a general trend towards less inclusion in glioma, the figure (Fig. 5c) does not explicitly show that glioma has less inclusion of microexons. Clarification of the x-axis and how dPSIs are calculated in this specific plot are required to address this discrepancy.

This data is now shown in Figure 5b. The y axis indicates the count of events with or without significant changes in human and/or mouse. We separated those counts according to the different combinations of change or lack thereof: blue for $dPSI < -0.1$, red for $dPSI > 0.1$, and gray for no change. dPSI values are calculated as the difference of PSI values in the two conditions. The plot shows that when the events are significantly changing in both species, the events tend to go in the same direction in both species, i.e. the first two bars are larger than the rest.

Additionally, the plot shows that in the particular case of microexons, the first bar (blue in both, i.e. less inclusion in glioma) is much larger than the second bar (red in both, i.e. more inclusion in glioma) or than any other possible combination. To further clarify this result, we have added a Figure (new Figure 5c) to make more explicit the direction of change of the microexons in human and mouse. This figure depicts the changes in translation of the conserved microexons and shows more explicitly the pattern of conservation described.

Reviewer #2 (Remarks to the Author):

Summary

Reixachs-Sole et al have developed a new pipeline to better understand translation of mRNAs at the level of mRNA isoform rather than gene. They have gone to validate the results of using this in both human and mouse, with a variety of different data sets and approaches. By comparing differences in mRNA isoform abundances and translation they show that several micro-exon containing isoforms are regulated between glioma and glioma.

The aim to understand translation at the mRNA isoform level is admirable and represents an important step forward in linking mRNA processing and translation. Overall, I think this is a good study but many aspects could benefit from improved explanations and examples to illustrate, especially for a general interest journal such as Nature Communications. **The focus ends up being on differential expression of microexons, between glioma and glioma, rather than differences in isoform abundance and translation, which seems the logical requirement for this new pipeline the authors developed here.** This work is novel and is certainly interesting to the gene expression field.

Specific points;

a) Authors mention that exon boundaries are frequently bound by RNA-binding proteins. However, the majority of these exon boundaries are at exon-intron boundaries in the nucleus. There is limited evidence to suggest that in spliced transcripts that make it out to the cytoplasm are more bound by RBPs than other parts of spliced transcripts.

We did not mean to say that there are more RBPs bound in the cytoplasm. We have eliminated this statement as it was not clear enough.

b) The manuscript would benefit greatly from a more detailed explanation of the novel ORQAS method that the authors have developed. For example, it seems like the designation of Ribo-seq reads to isoforms is based on relative abundance of mRNA isoforms from RNA-Seq-is this true? Then these transcripts would be filtered based on whether corresponding ORFs make it through cut-offs for uniformity of ribosome profiling reads across the ORF and periodicity. My concern is that transcripts may fail to pass these two thresholds for reasons other than alternative ORF translation. Similar metrics have been used previously to define translation events, so it is not obviously why they pipeline presented here is able to deconvolute translation of ORFs from alternatively spliced isoforms.

We have tried to improve the explanation of the pipeline in the text to clarify all these points and have included an extended description of the advantages with the corresponding tests. The assignment of Ribo-seq reads de novo, without any prior information does not work well because Ribo-seq produces not as many reads as RNA-seq. Additionally, these are shorted and not as uniformly distributed. As a consequence, direct isoform quantification methods like Kallisto or Salmon with Ribo-seq reads does not work well. Ribomap uses the RNA-seq abundance as priors for the optimization algorithm to distribute Ribo-seq reads among the different isoforms.

Multiple previous analyses have shown that uniformity and periodicity are essential to establish the Ribosome activity on an ORF. We thus applied the same principle in each isoform ORF, with the crucial difference that the reads used for that calculation are only those assigned to the isoform. To support this principle, we calculated the validation by immunohistochemistry (IHC) of the translation prediction on single-ORF genes. These are 1005 genes with one single ORF annotated, and non-overlapping with the single-ORF genes used as positive controls (used in Fig. 1b). For these genes we calculated the proportion of cases that have evidence of protein expression from immunohistochemistry (IHC) experiments from the human protein atlas (THPA):

Single-isoform genes do not have any ambiguity in the assignment of reads to ORFs. In the plot, “Not evaluated” are ORFs that do not have sufficient RNA expression to be considered for quantification, whereas “Not translated” are ORFs that despite having enough RNA and Ribo-seq reads, they do not pass the periodicity and uniformity cut-offs. Ribomap would predict the latter as translated. However, as shown, they have very little evidence of translation from THPA.

In the manuscript we further validated the predictions made with ORQAS in various ways:

- 1) Enrichment of mass-spec peptides in regions that are unique in isoforms
- 2) Enrichment of conservation of translated isoforms between human and mouse
- 3) Enrichment in high polysomal fractions of the translated isoforms vs the ones not-translated.

We also considered that the conservation between human and mouse of the differential inclusion in ORF abundance of microexons constitutes further evidence supporting that ORQAS can determine translation at isoform level.

c) Uniformity of periodicity could be used as a cut-off since changes on frame could indicate inconsistency caused but translation of ORF from alternative transcript. I guess the question is whether periodicity is stable across the transcript. How is periodicity calculated? It could be that the value of periodicity is an average across the ORF, so it is already reflecting whether it is constant enough.

Periodicity is calculated as the proportion of all reads mapping to the ORF that correspond to a given frame. We calculated the periodicity in 3 different windows of ~100 nucleotides at the beginning (START), in the middle (MIDDLE) and at the end of each ORF, separating ORFs according to different lengths. We can observe that the periodicity is uniform in these three windows and comparable to the total periodicity (now shown in Supp. Fig. 5):

We also analysed the distribution of differences of periodicity for each of these 3 windows in each ORF with the total periodicity of the same ORF and the majority of them do not show changes higher than 0.1 (dashed line) (now shown in Supp. Fig. 5):

d) In the section “Ribosome profiling discriminates translation abundance at isoform level” it is not clear what “combination of protein features” means in Fig 2b. It seems unsurprising that isoforms called translated have evidence of their protein expressed. It would be informative to generate a false discovery rate, for those isoforms that have no evidence of translation. It is not clear how mass spec, immunohistochemistry and uniprot data was treated to ensure that signal could be confidently assigned to a specific ORF isoform over another?

Previous fig. 2b (now moved to Supp. Fig. 3) represents a validation at the gene level using the protein expression annotation from THPA, which is based on Mass Spectrometry, Immunohistochemistry, and Uniprot (associated known protein). The intention of this plot was to provide a first coarse-grained validation of predictions. For every gene with one or more translated ORFs predicted, we calculated whether there was evidence of translation for that gene. The plot shows that most of the genes for which we predict one more translated isoform has indeed some protein expression evidence (7992 out of 7992+365).

e) In the text, polysome association is described, whereas in Fig 2c, high polysomes are mentioned. What specific complexes were defined as high polysomes? How many transcripts were included in this monosome vs high polysome distribution? From RNA-Seq data it is not clear how we can be sure that this is transcript specific. If RT-qPCR with primers designed specifically for detection of specific isoforms match this same pattern?

Polysome fractions used in our analysis were defined in (Blair et al. Cell Reports 2017), which was based on the procedures described in (Floor & Doudna Elife 2016). Their definition was as follows: Monosomes = 1 ribosome, Low polysomes = 2-4 ribosomes, High polysomes = 5 or more ribosomes. In the table below we show the number of expressed transcripts in the analysed data:

Condition	Fraction	Expressed transcripts
hESC	High Polysome	57562
hESC	Monosome	48967

Neu	High Polysome	49342
Neu	Monosome	43023

We used Salmon to assign reads from each polysomal fractions to the transcript isoforms. Salmon performs an unambiguous assignment of reads to transcripts based on the similarities of the read to other reads mapping to the same transcript using an optimization. Abundances can be estimated to each transcript in each subpopulation of RNAs from the corresponding RNA-seq. The abundances were normalized as in (Maslon et al. Elife 2014) by dividing the abundance in a given fraction over the total abundance in all fractions being compared. In general, these normalized values are never 0 (does not appear in fraction) or 1 (unique to the fraction), but the values vary enough to determine the enrichment in specific fractions, as shown before in (Maslon et al. Elife 2014).

f) The validation performed in Fig3a is excellent. But these sections would benefit greatly from examples illustrating these types of events.

We have generated plots for isoform-specific regions with uniquely-mapped reads for a couple of examples: ENSG00000196867 and ENSG00000213995, which are now shown in Supp. Fig. 11 and Supp. Fig. 12. In these plots we show the uniquely-mapped Ribo-seq reads (green), Ribo-seq reads that map to 2 different places (red), and Ribo-seq reads that map to 3 different places (blue).

g) There is very little explanation of many of the panels. For example, 3c): what was the aim of this over how mass spec data had be used to support ORF isoform translation?

We have tried to improve the description in the legend and in the text of the various figures. In previous Fig. 3c, Mass-spec peptides from mouse hippocampus and glia were used to validate unique regions. These are two types of regions: those sequences that are specific to a single isoform (unique sequence) and those ORFs that are specific to an isoform (the sequence might be shared with other isoforms, but the ORF is unique and specific to the isoform). We have modified the cartoon in Figure 3a to clarify the possible configurations of these regions. For improved clarity, we have also separated into independent panels the validation of isoform-specific sequences with Ribo-seq reads (now in Figs. 3b and 3c) and the validation of isoform-specific ORFs with P-sites or mass-spec peptides (now in Figs. 3d and 3e).

h) It is not clear how S4 is different to Fig3? Is it same but just including human samples too?

Supp Figure 4 (now Supp. Fig. 13) represents the same type of analyses as Fig. 3 (now in Figs. 3b, 3c, 3d and 3e) but for other samples. These figures represent the density of Ribo-seq reads in isoform-specific sequences (Fig. 3b), isoform-specific sequences with 10 or more Ribo-seq reads (Fig. 3c), the density of P-sites in isoform-specific ORFs (Fig. 3d) and the number of isoform-specific ORFs with one or more peptides from Mass-Spec. Supp. Fig. 13 contains the same plots for the samples not included in Fig. 3.

i) Use of SUPPA in Fig4 to probe differential translation linked to differential splicing is really the most interesting part of the manuscript. Since one of the big questions in the field is whether certain spliced isoforms are preferentially translated. This analysis starts to address this. Fig 4c suggests this is generally not the case. The majority of analysis focuses on whether the translation of spliced microexons and their differential splicing. This is an important question and result. However, it is not clear whether this analysis was dependant on the original pipeline ORQAS, developed here.

Yes, we used ORQAS to quantify ORFs in ribosome space, and then used SUPPA to transfer that information at event level. We have tried to explain this better in the manuscript.

j) The focus of results seemed to be on changes correlating between RNA-Seq and Ribo-Seq, especially in Fig 5, that are differential between different cell types. But given the manuscript aims to understand isoform translation it would be more appropriate to analyse more deeply events whose RNA-Seq and Ribo-Seq don't correlate. These are the situations, one would argue, that require understanding of isoforms at both splicing and translation level.

Our main objective is to estimate the impact of differential splicing on translation. We argue that to be able to address that one needs to evaluate isoform translation, since differential expression and splicing is simply a result of what happens to the isoforms. Those isoforms whose RNA-seq abundance and Ribo-seq abundance do not correlate are probably related to translation efficiency (TE), which is out of the scope of this manuscript.

We considered those events that changed significantly in opposite directions in RNA and Ribosome space. We expect those that increase significantly PSI in RNA but decrease in Ribosome would be related to a decrease in TE, and those events that decrease significantly in PSI in RNA but increase in Ribosome would be related to an increase in TE. We calculated the TE in these cases:

Except for a few cases, the TE is similar in glioma and glia for all isoforms. This suggests that TE does not change globally and explains a very limited number of significant changes in splicing and translation.

k) In discussion “These estimates are far from” should be reworded to give indication of direction of change.

We have changed it to “considerably lower than”

Reviewer #3 (Remarks to the Author):

The authors have developed a new method, ORQAS (ORF quantification pipeline for alternative splicing) to quantify isoform-specific translation abundance, and have applied their method to a number of different datasets including glioma/glioma and ES cells/neurons.

One of the limitations of ribosome profiling is that it does not directly measure protein peptide abundance and assumes that engaged ribosomes are direct arbiters of protein levels. As the authors point out, this is not strictly true, and builds upon previous studies to attempt to examine open reading frames that are more likely to be translated. Their polysome data is convincing.

While I have some concerns about the novelty given that other studies examining alternative splicing and ribosome profiling have previously been published, I do think that the authors have significantly improved upon these studies and this study deserves to be published in Nature Communications.

I have a few queries:

1. In Figure 4C, the correlation between the ribo-seq and RNA-seq data is remarkably high, close to 1 - almost too good to be true. What happens to the datasets that are not significant, i.e.. what is the correlation between non-significant alternative splicing events for which there is ribo-seq data and vice-versa?

We have plotted now all the events in this comparison (now shown in Fig. 4b for human and in Supp. Fig. 14c for mouse):

In these figures we show in red the cases that are significant in both Ribo-seq and RNA-seq space. In black or dark gray, we show the cases that are only significant in one case, and in light gray, the cases that are significant in neither of them. The density bands in the plot shows the distribution of cases along the axes. Empty circles indicate the microexons (red if significant, black/dark-gray/gray otherwise). In the inset in blue we give the correlation of the red points (including microexons), in gray we give the correlation of all other exons (black/dark-gray/gray, including microexons). There is overall a considerably high correlation of the non-significant cases (dark/dark-gray/gray points), but the correlation of the significant cases is higher.

2. I would suggest that the authors improve their description of their method in the first results section, including on expanding on the descriptions of uniformity and periodicity and why the integration of which is a significant advance over ribosome profiling alone.

We have added more details about the method in the first section. We have also added new analyses to justify the different steps in the method and we show a new analysis to show that the periodicity does not across ORFs. Among the tests to better explain the advantages of the ORQAS we selected genes with one single ORF annotated (different gene set from the one used as positive control in Fig. 1b) and calculated the proportion of cases that have evidence of protein expression from immunohistochemistry (IHC) experiments from the human protein atlas (THPA):

The cases labelled “Not evaluated“ are those that do not show enough RNA expression to be evaluated by ORQAS. The cases labelled “Not translated“ are the predictions that Ribomap would give without any filters for periodicity or uniformity. In both cases there was an enrichment of cases without evidence of translation from TPHA. In contrast, after imposing the thresholds (the cases labelled “Translated“), we observed a higher proportion of cases with protein evidence at different levels. We show this now in Figure 1c. To further show that ORQAS is superior to other methods, we have included an exhaustive comparison with SaTAnn and with the results from (Weatheritt et al. 2016). These comparisons show that ORQAS can detect more translated alternative isoforms per gene, less potential false positives, and more microexons.

REVIEWERS' COMMENTS:

Reviewer #1 (Remarks to the Author):

In our first review, our major concerns were that the paper lacked a comparison with other methods (e.g. RiboMap and SaTAnn), testing of the uniformity and periodicity parameters, and additional validations such as quantification between cytoplasmic and nuclear extracts.

The authors adequately answered our concerns by:

- Testing if one single ORF genes (1005) had protein expression evidence (according to immunohistochemistry experiments from the human protein atlas), as well as being better detected by ORQAS than with RiboMap
- Comparing a Weatheritt et al 2016 dataset with ORQAS and running SaTAnn with the same samples.
- Comparing the capacity of ORQAS and SaTAnn in detecting short unique regions and microexon-containing isoforms.
- Comparing the ORF length in high-polysomal and monosomal fractions.
- Testing the robustness of ORQAS by comparing the expression of tissue-specific genes (according to TPHA) and calculating translation in their glial samples.

We would have preferred if the authors had made a simulated dataset of Ribo-seq and used it to compare ORQAS, RiboMap and SaTAnn. Nevertheless, the authors nicely show that ORQAS is able to quantify transcript isoforms using Ribosome profiling reducing the number of false-positive results in comparison with previous methods.

Minor concerns:

We also asked that they should show the exact p-values in the figures and text, we request this should be done for easier interpretation. As well, we request that all figures and text should have the same consistency throughout the document showing the exact p-values and statistical tests used.

We found that some minor corrections in the new data such as:

- Fig. 3f and Supp Fig. 13a the names of the x-axis are missing. We cannot tell which data is of human or mouse samples.
- In the response to referees, regarding the section discussing translation in monosomes and polysomes, the authors mention that they did not see any significant difference in the distribution of transcript lengths between monosomes and high-polysomes and show the distribution of ORF length of monosomes and high-polysomes. We ask the authors to test the homoscedasticity of these distributions, as it seems they are bimodal.
- In the methods, they should explain the specific parameters used in SaTAnn.
- All legends need to state the number of data points used in the figure.
- The authors should also articulate the limitations of their method, for example, the use of periodicity means that events in the 5' UTR are not evaluated.

Reviewer #3 (Remarks to the Author):

The authors have done a thorough job addressing the reviewers concerns and the manuscript is much improved as a result. I am happy to recommend publication of this manuscript and congratulate the authors on a great study.

Reviewers' comments:

Reviewer #1 (Remarks to the Author):

In this manuscript, the authors describe their new pipeline for open reading frame (ORF) quantification for alternative splicing, ORQAS. It is able to quantify the translation of individual transcript isoforms using Ribosome profiling data. They address the challenge that detection of protein variation derived from differential microexon using unbiased proteomics is nowadays not possible, whereas their pipeline is able to detect conserved microexons between human and mouse. The highlight of their pipeline is how they determine the potential of ORF translation, by calculating the uniformity and the periodicity along the ORF. **This paper lacks comparison with other computational methods (e.g RiboMap and SaTAnn), testing of the uniformity and periodicity parameters, and additional validations such as quantification between cytoplasmic and nuclear extracts) to be considered as a novel method for alternative splicing quantification. Furthermore, the description of many of the figures makes interpretation of the results difficult to evaluate and the methods section requires more extensive description.**

Algorithm outline

- 1) Quantifies the abundance of ORFs in from RNAseq (TPMs).
- 2) Assigns ribosome sequencing (Ribo-seq) to the same ORFs by RiboMap.
- 3) Calculates for each ORF two parameters for determining their potential translation: Uniformity (proportion of the maximum entropy of the read distribution) and the 3nt signal periodicity along the ORF.

Major concerns

- 1) The authors ORQAS algorithm relies primarily on a published pipeline from the Kingsford group (Wang et al. 2016, Bioinformatics). The innovation they've applied is to integrate in uniformity and periodicity cut-offs. However, they provide no evidence that these measures are an improvement over the approach used by Ribomap. Since the authors are releasing their own package, ORQAS, they should demonstrate the unique strength and advantage of their approach.

To show that our approach provides an advantage over the approach used by Ribomap, we performed the following comparison. We selected genes with one single ORF annotated, so there is no ambiguous mappings of Ribo-seq reads. These are 1005 genes with one single ORF annotated, and non-overlapping with the single-ORF genes used as positive controls. For these genes we calculated the proportion of cases that have evidence of protein expression from immunohistochemistry (IHC) experiments from the human protein atlas (THPA):

The cases labelled “Not evaluated” are those that do not show enough RNA expression to be evaluated by ORQAS. The cases labelled “Not translated” are the predictions from Ribomap without any filters for periodicity or uniformity. In both cases there was an enrichment of cases without evidence of translation from THPA. In contrast, after imposing the thresholds (the cases labelled “Translated”), we observed an enrichment of cases with protein evidence at different levels. This is now shown in Figure 1c.

Additionally, to show the unique strength and advantage of our approach, we calculated the proportion of translated isoforms according to the minimum RNA expression cut-off, which is another one of our filters. We observed that the proportion of translated isoforms increases as a function of the expression cut-off:

We have added this plot as supplementary Figure.

To further show the robustness of ORQAS, we also generated plots with the proportion of translated isoforms validated by peptides as a function of the RNA isoform expression cut-off. They all increase with higher expression cut-offs. Thus, using a higher expression cut-off translation can be validated with greater confidence:

We have also included this plot as supplementary Figure 13.

2) In addition, the authors need to show that their cutoffs for uniformity and periodicity improve their ability to reliably detect active translation and that they are not removing data unnecessarily.

As shown above, the cutoffs of uniformity and periodicity improve the ability to detect cases with evidence of translation. We have added these plots to the article and have included an explicit mention to this in the text (highlighted in blue).

3) Previous work both published (Weatheritt et al. 2016, Sterne-Weiler et al. 2014, Floor and Doudna. 2016) and in prepublication servers (Calviello et al.

<https://www.biorxiv.org/content/10.1101/608794v1>) have identified alternative splicing variants engaged with polysomes or ribosomes. Furthermore, both Weatheritt et al. and Calviello et al. have analyzed ribosome profiling data and have provided estimates for the number of splice variants engaged by the ribosome. The authors explain the advantages of their approach but do not contrast their results, especially in light of controversy of splice variant identification in mass spec data.

In the original submission we cited (Weatheritt et al. 2016), (Sterne-Weiler et al. 2014), and (Floor and Doudna. 2016). The preprint Calviello et al. appeared after our preprint <https://www.biorxiv.org/content/10.1101/582031v2> and also after we submitted our work for peer review.

In the original submission we used data from (Blair et al. 2017) for human ESCs and differentiated neuronal cells to validate some of our findings for microexons. We also used the mapping of RNA sequencing reads from polysomal fractions, which is the approach of (Sterne-Weiler et al. 2014), as independent validation of our findings.

In the revised version we now include an exhaustive comparison with the data from Weatheritt et al. 2016 and with the method SaTAnn (see below).

In our manuscript we described the differences of our method with previous approaches (Weatheritt et al.): ORQAS estimates the engagement of ribosomes in transcript variants, measuring for each transcript isoform independently their periodicity and uniformity. In contrast Weatheritt et al. mapped Riboseq reads directly to exon-exon junctions without considering whether their validity in terms of the transcript-isoform level expression, and Ribo-seq uniformity or periodicity. We argue that, since ribosomes scan transcript molecules, ORQAS provides a description that is easier to interpret in terms of translation of an mRNA isoform and also one that is closer to what is actually happening in the cell. SaTAnn considers directly the P-sites from Ribo-seq reads per exon, and combines exons into transcripts, without considering the RNA expression. In the revised manuscript we show that the

translation quantification provided by ORQAS and SaTAnn is similar for isoforms predicted by both methods, but SaTAnn predicts translation also in isoforms that show no RNA expression (see below for more details of this comparison).

Another novelty of ORQAS with respect to previous studies is that we compare with mouse and find that the direction in which alternative splicing potentially impact translation is conserved in specific events, thereby establishing the potential functionality of these changes.

4) We undertook a brief comparison to SaTAnn, the authors should expand on this. SaTAnn versus ORQAS. Both approaches used HEK293 data (<https://www.nature.com/articles/nmeth.3688> and GSM1306496)

ORFs with quantified translation Number of genes Number of genes with only one ORF translated
 Number of genes with translation of multiple ORFs
 SaTAnn ~24,000 ~15,000 9,138 (60.92%) >5,800
 ORQAs ~20,709 Not explicitly mentioned 5,237- 4,766
 (Fig 1c -1d) 52.3-54.9%
 (Fig 1c and 1d)

SaTAnn operates differently from ORQAS. SaTAnn works directly with P-sites, without taking into account RNA-seq abundances, it analyses first exons, which are then combined together. We have performed an exhaustive comparison with SaTAnn. To this end, we run SaTAnn with the same samples we analyzed in our manuscript. We have added these analysis to the manuscript.

We first compared the number of genes predicted to have at least 1 translated ORF (left panel below) and the total number of translated ORFs predicted (right panel below). We do not observed any systematic trend in these results. In some samples, ORQAS predict more translated genes and a more translated ORFs, and in other samples the trend is the opposite:

Interestingly, although both methods predict a similar number of genes in human glioma (hsa glioma), ORQAS predicts a larger number of ORFs. Also, although SaTAnn predicts more genes with translated ORFs in mouse glia (mmu glia), ORQAS predicts more ORFs overall. This suggests a difference in the number of translated ORFs per gene. To investigate this further, we calculated the

number of genes with 1, 2, 3,... translated isoforms. We observed that ORQAS detects more genes with multiple ORFs:

This is now shown in Fig. 1f and in Supp. Fig. 6. We further calculated the agreement between both methods in terms of the genes translated and the total ORFs using a Jaccard Index = $\text{Intersection(ORQAS, SaTAnn)} / \text{Union(ORQAS, SaTAnn)}$. We observed a high level of agreement at gene level (60-80%) (left panel below), but slightly lower level of agreement at isoform-ORF level (right panel below) (now shown in Supp. Fig. 7):

We also compared the quantification provided by both methods. SaTAnn provides a quantification based on the normalized number of P-sites per nucleotide (y axis below), whereas ORQAS provides a quantification in ORFs per million (OPM) (x axis below), akin to the TPM units. We observed that for

the ORFs predicted as translated by both methods, the quantification values correlate ($R=0.8076$) (all plots now shown in Fig. 1g and Supp. Fig. 8):

However, there were a number of cases (in green in the figure above) that ORQAS did not predict because they had not sufficient reads or no reads at all, but are translated and with high abundance value according to SaTAnn. Looking further at these differences, we observed that the ORFs predicted by SaTAnn that were not detected by ORQAS had low or no RNA-seq expression (plots now shown in Fig. 1h and Supp. Fig. 8):

Thus SaTAnn predicted ORFs as translated even if they had no evidence of RNA expression. Considering how short Ribo-seq reads are, it is possible that these cases originate from incorrect mapping and are thus false positives. Cases with too low or no RNA expression evidence are discarded by ORQAS, so these potential false positives are avoided.

We additionally tested the capacity of ORQAS and SaTAnn to detect short unique regions, i.e. regions that are specific to the isoform ORF. We observed that ORQAS is able to detect more ORFs with short unique regions (Now shown in Supp. Fig. 15):

The plot above shows the number of translated ORF predicted (y axis) as a function of the length of their unique regions (x axis). Thus ORQAS provides an advantage to predict translation in ORFs with short unique regions.

We also analysed the capacity of ORQAS and SaTAnn to detect microexon-containing isoforms, defining microexons as exons of length 51nt or shorter (more on this below). ORQAS recovered a larger number of microexon-containing isoforms across the different samples tested (now shown in Fig. 4f):

We also performed a comparison with the data from Weatheritt et al. 2016. To perform this comparison, we run ORQAS with the same Hek293 sample used in Weatheritt et al. 2016 and took all the events that were potentially measurable by both methods. From these events, we plot below those that were detected (PSI>0 in Ribosome space) by both methods, or by each method independently (now shown in Fig. 4g):

We also calculated the expression in Ribosome space (OPM units) and in RNA space (TPM units) of the genes where SE events were detected. Weatheritt et al. also finds events for which we cannot find expression:

This highlights once again the potential artefacts that may appear by mapping Ribo-seq reads directly to the alternative splicing events, and the advantage of ORQAS approach to avoid these potential false positives.

5) In the section: Ribosome profiling discriminates translation abundance at isoform level. The authors' validate their approach by comparing translated isoforms predicted by ORQAs with RNAseq of polysomal fractions and found that their predictions are enriched in the polysomal fraction but not in monosomes (Fig 2c). This validation is not sufficient to support the translation prediction power of ORQA as active translation can also take place in monosomes (See Heyer and Moore, 2016). We propose they should compare between cytoplasmic and nuclear extracts.

Similar to translation on monosomes, there is also evidence of translation in the nucleus: <http://jcb.rupress.org/content/197/1/45>, so it is not entirely clear whether a comparison between genes expressed in the nucleus and cytoplasm will be necessarily conclusive.

Although translation can indeed occur in monosomes, we expect that mRNAs translating in monosomes should be generally short, as they would only hold 1 individual ribosome. First, we did not see any significant difference in the distribution of transcript lengths between monosome and high-polysome fractions in general (left plot below). We then separated between the cases that we defined as translated and not-translated. We observed that translating mRNAs are significantly shorter in monosomes (right plot below). This difference is not as large when considering not translated ORFs.

This comparison is now shown in Supp. Fig. 10. We thus separated previous Figure 2c according to three length ranges and observed that there was the same trends as before for translated transcripts in each of the length ranges (Now shown Fig. 2a):

As an additional test to show the robustness of ORQAS in defining translation at isoform level, we considered tissue specific genes as proxy for negative controls in a different tissue. We took genes annotated in THPA to have tissue specific expression in brain, heart, intestine, liver, spleen or testis, and calculated whether they were predicted as translated or not in our glia samples (now shown Supp. Fig. 3):

ORQAS on glia data predicts a higher proportion of translated genes in the subset of brain-specific genes compared with the subsets of genes specific in the other tissues.

6) The authors need to provide a more detailed methods section. For example, we advise that the authors elaborate further on how they use SUPPA to convert isoform abundances to event inclusion values together with rationales that validate such a conversion strategy.

We have tried to explain this better in the text. This conversion is supported by previous validations with RNA (see Alamancos et al. 2015, Trincado et al. 2018). With RNA, the abundance values from transcript isoforms summarized per event as a relative abundance (PSI) agrees with the measurements from RT-PCR using probes to capture the relative abundances for that event. For Ribo-seq we reasoned that, if Ribo-seq can be used to estimate the translation abundance of an isoform, we can also summarize these values per event and calculate a relative translation abundance. This represents the relative contribution of that particular exon to the translation abundance from a set of isoforms. That is, the relative contribution from an alternative exon to the translation of a gene.

7) The microexon section is interesting, especially the identification of microexons in glia samples. However, the majority of previous studies have used a cutoff of <30 nt for microexons (Irimia et al, 2014; Torres- Méndez et al, 2019). Does the enrichment still exist with this cutoff?

We confirmed our observations with exons of length <28nt by testing their enrichment in the set of alternative exons changing inclusion in RNA or Ribosome space between glia and glioma, and between neural and embryonic sample (now included in the text):

Comparison	Sequencing	Species	Fisher test p-value
Glia vs glioma	RNA-seq	human	5.435e-13
Glia vs glioma	Ribo-seq	human	1.17e-09
Glia vs glioma	RNA-seq	mouse	7.47e-14
Glia vs glioma	Ribo-seq	mouse	3.194e-06
hESC vs neural	RNA-seq	human	2.725e-08
hESC vs neural	Ribo-seq	human	6.768e-06

Irimia et al. 2014 justified the definition of microexon as <28nt from their observation that below this length there was an increased inclusion in neuronal samples with respect to non-neuronal samples. However, other length cut-offs have been used to define microexons (see e.g. Li et al. 2015, Ustianenko et al. 2017). To justify our choice of the <52nt cut-off, we looked at the distribution of inclusion levels (in PSI units) for exon-cassette events in RNA and Ribosome space, separated by exon-length ranges in glia and in glioma (now shown in Supp. Fig. 14e):

This plot shows that at length range [1,27] the differences of inclusion between glia and glioma is the largest, as expected for the definition of microexon from (Irimia et al. 2014). However, in the range (27,51] there is also a difference in inclusion between glia and glioma, which would support the inclusion of these exons into the definition of short exons with a brain cell specific inclusion pattern, or microexons.

Minor concerns

- TPM value of 0.1 is very low. We suggest an alternative cutoff of at least 1 for robust interpretation.

Using bins for different range of expression cut-off values, we observed that the higher the minimum expression considered, the larger the proportion of cases that are predicted as translated (now shown in Supp. Fig. 2):

However, the proportion of transcripts predicted as translated does not change much after removing cases below 1 TPM.

- Fig 1b. Legend says: single ORF housekeeping genes are in blue but in the figure it suggests otherwise.

We have fixed the figure legend.

- In the section: Ribosome profiling discriminates translation abundance at isoform level
 - 1) The authors mention they have 15,824 human-mouse 1-1 gene orthologs, and identify 18,574 human-mouse protein isoform pairs representing functional orthologs. They then state to find 7,112 (64%) of the 1-1 gene orthologs had more than one orthologous isoform pair. We suggest the authors elaborate on how they reached this figure, as well as provide more information about the relevant plot.

We have provided more information in the manuscript about this calculation and about the figures shown. Orthology is only annotated in databases at the gene level. However, we needed to obtain the pairs of potential ORF orthologs to establish conservation. We thus selected first the set of 15,824 1-1 gene orthologs between human and mouse, i.e. best reciprocal orthology assignments between human and mouse, hence no ambiguous mappings. We then needed to establish the orthology at protein isoform level, i.e. the proteins in each pair of genes that can be considered protein orthologs. For that we considered all pairwise global alignments between human and mouse proteins, and selected the best possible pairs above a minimum score of 0.8 (defined as the fraction of amino acid matches over the total length of the global alignment), using a symmetric version of the stable

marriage algorithm, as described before (Eyras et al. 2004). In this algorithm, pair assignments are established given an ordering of “preference” for each element, which is provided by the score calculated from the alignment. The algorithm produces optimal or “stable” pairs, in the sense that it provides the best available matches rather than the best reciprocal matches. That is, given a stable pair, there are no other pairs possible where both elements would prefer each other more than their current pair. This algorithm produces 18,574 human-mouse protein isoform pairs. We calculated then how these are distributed in genes orthologs: 36% of gene orthologs had one single protein isoform pair, and 64% had 2 or more protein isoform pairs.

2) In the text, the p-value is as “< 2.2 e-16”, whereas in the figure legend they show the exact p-values.

This was to avoid writing all p-values in the text and to indicate that all tests were significant and with p-values smaller than 2.2e-16. The text says “... were significantly enriched in translated isoforms in both species (p-value < 2.2e-16 in all datasets)”. The figure legend shows the actual p-values.

- Figures 3b and Supp. Fig 4a show the density of reads per nucleotide compared with other isoforms and suggest that “both region types in translated isoforms showed a higher density of reads per nucleotide compared with other isoforms”
 - We believe these figures do not necessarily agree with the aforementioned statement from the main text and require further elaboration in the Figure legends such as:
 - Clarification of whether the specific sequence sets are independent.
 - Present results of statistical test (eg. Wilcoxon rank sum test) for significant difference.
 - We also suggest the authors present the data with Ecdf plot to contrast the distributions between datasets.

We considered isoform-specific regions, since evidence mapped to these regions can then be unequivocally assigned to the isoform. We defined two types of isoform-specific regions. One type was defined in terms isoform-specific nucleotide sequences, i.e. continuous nucleotide stretches that are only included in an isoform. To validate our predictions with peptides from MS experiments and P-sites, we additionally considered isoform-specific ORF regions. These were defined as sequences that may or may not be shared between isoforms but had a specific frame in each isoform, so that peptides from MS experiments can be unequivocally mapped on these regions. These regions included the isoforms calculated before with the ORFs from the isoform-specific sequence.

Isoform-specific sequences are thus also isoform-specific ORFs, e.g. an alternative exon specific to an isoform defines an isoform-specific ORF region. Thus the two sets of isoforms are not independent. However, the datasets used in each case are independent. Isoform-specific ORFs were validated with P-sites and with mass-spec peptides, whereas isoform-specific sequences were validated with Ribo-seq reads, regardless of the position and identification of the P-site. We have tried to clarify this in the text, and added new plots. Significance is now shown in new plots in Figure 3 and in Supp. Fig. 13. Below we show the case for the densities of Ribo-seq reads in isoform-specific sequences:

We have modified the cartoon in Figure 3a to clarify the possible configurations of the isoform-specific regions. For improved clarity, we have also separated into independent panels the validation of isoform-specific sequences with Ribo-seq reads (now in Figs. 3b and 3c) and the validation of isoform-specific ORFs with P-sites or mass-spec peptides (now in Figs. 3d and 3e).

- It is unclear how Figure 3e and 3f support statements from the main text. In particular, the statement of “63-65% in mouse with an RNA expression > 0.1 TPM” is difficult to relate to the data presented in Figure 3 without further elaboration. In addition, the authors should provide further information Figure 3f supporting the statement that “~ 10% of annotated AS in both human and mouse had evidence of translation and these represented 60% of all translated isoforms”, as unclear how these numbers were reached.

Figure 3e is now shown in Fig. 3f, and Fig. 3f has been moved to Supp. Fig. 13e for clarity. Fig. 3f (previous Fig. 3e) shows the proportion of all expressed isoforms that are predicted to be translated and additionally have independent validation of translation. This validation is considered to be one or more of different sources of evidence: conservation, uniquely mapped Ribo-seq reads in isoform-specific sequences, and counts per base or peptides in isoform-specific ORFs. Supp. Fig. 13e (Previous Fig. 3f) shows the same information but splitting isoforms according to whether they are the main isoform or an alternative isoform, where “main” was defined as the isoform with the highest expression, and “alternative” are the rest of isoforms from the gene showing expression (TPM>0.1). The proportions shown were calculated by putting together all the results of the validation analyses and are available in Supp. Tables 3 and 4.

- Fig. 4a) in the section of RA calculation, the figure says “RA = OPM1 + OPM2”. We believe that the authors' initial intention being to calculate RA=OPM1 + OPM3. If so, please correct this typo.

That's right. We have corrected this error.

- Fig 4e and 4f are not sufficient to support a direct connection of microexons included in RNAseq to Ribo-seq. Inclusion of a plot supporting a strong correlation between RNAseq and Riboseq for microexons (as Figure 4c) is strongly advised.

We have included now in Figure 4b (and the Supplementary Figure for 14c mouse) also the cases that are not significant in RNA-seq or Ribo-seq or both in the comparison between glia and glioma, also highlighting (as empty circles) microexons:

The figure shows in red the cases that are significant in both Ribo-seq and RNA-seq space, in black or dark gray, the cases that are only significant in one case, and in light gray, the cases that are significant in neither of them. The density bands in the plot shows the distribution of cases along the axes. Empty circles indicate the microexons (red if significant, black/gray otherwise). In the inset in blue we give the correlation of the red points (including microexons), in gray we give the correlation of all other exons (black/dark-gray/gray, including microexons).

- Fig 5b The authors mention: "high proportion of them changed in the same direction between glioma and glioma (66% in RNA-seq and 78% in Ribo-seq)". The authors need to provide more information, as this result cannot be clearly inferred from Fig.

We have provided more information in the text to clarify this result. Additionally, for clarity we have put together Figures 5b and 5c (now Fig. 5b) and have improved the figure caption to make it more clear. In these barplots we show the events changing significantly in human and/or mouse in each direction: less inclusion (blue), more inclusion (red). We have also added a new plot (now Figure 5c) comparing the difference of inclusion values in human and mouse in Ribo-seq space, which shows that microexons have a conserved pattern of decreased inclusion in glioma. More details on this below.

- Similar to the previous comment, the authors mention that microexons were enriched in both species with a general trend towards less inclusion in glioma, the figure (Fig. 5c) does not explicitly show that glioma has less inclusion of microexons. Clarification of the x-axis and how dPSIs are calculated in this specific plot are required to address this discrepancy.

This data is now shown in Figure 5b. The y axis indicates the count of events with or without significant changes in human and/or mouse. We separated those counts according to the different combinations of change or lack thereof: blue for dPSI < -0.1, red for dPSI > 0.1, and gray for no change. dPSI values are calculated as the difference of PSI values in the two conditions. The plot shows that when the events are significantly changing in both species, the events tend to go in the same direction in both species, i.e. the first two bars are larger than the rest.

Additionally, the plot shows that in the particular case of microexons, the first bar (blue in both, i.e. less inclusion in glioma) is much larger than the second bar (red in both, i.e. more inclusion in glioma) or than any other possible combination. To further clarify this result, we have added a Figure (new Figure 5c) to make more explicit the direction of change of the microexons in human and mouse. This figure depicts the changes in translation of the conserved microexons and shows more explicitly the pattern of conservation described.

Orthologous SE events in Ribo-seq

Reviewer #2 (Remarks to the Author):

Summary

Reixachs-Sole et al have developed a new pipeline to better understand translation of mRNAs at the level of mRNA isoform rather than gene. They have gone to validate the results of using this in both human and mouse, with a variety of different data sets and approaches. By comparing differences in mRNA isoform abundances and translation they show that several micro-exon containing isoforms are regulated between glioma and glioma.

The aim to understand translation at the mRNA isoform level is admirable and represents an important step forward in linking mRNA processing and translation. Overall, I think this is a good study but many aspects could benefit from improved explanations and examples to illustrate, especially for a general interest journal such as Nature Communications. **The focus ends up being on differential expression of microexons, between glioma and glioma, rather than differences in isoform abundance and translation, which seems the logical requirement for this new pipeline the authors developed here.** This work is novel and is certainly interesting to the gene expression field.

Specific points;

a) Authors mention that exon boundaries are frequently bound by RNA-binding proteins. However, the majority of these exon boundaries are at exon-intron boundaries in the nucleus. There is limited evidence to suggest that in spliced transcripts that make it out to the cytoplasm are more bound by RBPs than other parts of spliced transcripts.

We did not mean to say that there are more RBPs bound in the cytoplasm. We have eliminated this statement as it was not clear enough.

b) The manuscript would benefit greatly from a more detailed explanation of the novel ORQAS method that the authors have developed. For example, it seems like the designation of Ribo-seq reads to isoforms is based on relative abundance of mRNA isoforms from RNA-Seq—is this true? Then these transcripts would be filtered based on whether corresponding ORFs make it through cut-offs for uniformity of ribosome profiling reads across the ORF and periodicity. My concern is that transcripts may fail to pass these two thresholds for reasons other than alternative ORF translation. Similar metrics have been used previously to define translation events, so it is not obvious why the pipeline presented here is able to deconvolute translation of ORFs from alternatively spliced isoforms.

We have tried to improve the explanation of the pipeline in the text to clarify all these points and have included an extended description of the advantages with the corresponding tests. The assignment of Ribo-seq reads de novo, without any prior information does not work well because Ribo-seq produces not as many reads as RNA-seq. Additionally, these are shorted and not as uniformly distributed. As a consequence, direct isoform quantification methods like Kallisto or Salmon with Ribo-seq reads does not work well. Ribomap uses the RNA-seq abundance as priors for the optimization algorithm to distribute Ribo-seq reads among the different isoforms.

Multiple previous analyses have shown that uniformity and periodicity are essential to establish the Ribosome activity on an ORF. We thus applied the same principle in each isoform ORF, with the crucial difference that the reads used for that calculation are only those assigned to the isoform. To support this principle, we calculated the validation by immunohistochemistry (IHC) of the translation prediction on single-ORF genes. These are 1005 genes with one single ORF annotated, and non-overlapping with the single-ORF genes used as positive controls (used in Fig. 1b). For these genes we calculated the proportion of cases that have evidence of protein expression from immunohistochemistry (IHC) experiments from the human protein atlas (THPA):

Single-isoform genes do not have any ambiguity in the assignment of reads to ORFs. In the plot, “Not evaluated” are ORFs that do not have sufficient RNA expression to be considered for quantification, whereas “Not translated” are ORFs that despite having enough RNA and Ribo-seq reads, they do not pass the periodicity and uniformity cut-offs. Ribomap would predict the latter as translated. However, as shown, they have very little evidence of translation from TPHA.

In the manuscript we further validated the predictions made with ORQAS in various ways:

- 1) Enrichment of mass-spec peptides in regions that are unique in isoforms
- 2) Enrichment of conservation of translated isoforms between human and mouse
- 3) Enrichment in high polysomal fractions of the translated isoforms vs the ones not-translated.

We also considered that the conservation between human and mouse of the differential inclusion in ORF abundance of microexons constitutes further evidence supporting that ORQAS can determine translation at isoform level.

c) Uniformity of periodicity could be used as a cut-off since changes on frame could indicate inconsistency caused but translation of ORF from alternative transcript. I guess the question is

whether periodicity is stable across the transcript. How is periodicity calculated? It could be that the value of periodicity is an average across the ORF, so it is already reflecting whether it is constant enough.

Periodicity is calculated as the proportion of all reads mapping to the ORF that correspond to a given frame. We calculated the periodicity in 3 different windows of ~100 nucleotides at the beginning (START), in the middle (MIDDLE) and at the end of each ORF, separating ORFs according to different lengths. We can observe that the periodicity is uniform in these three windows and comparable to the total periodicity (now shown in Supp. Fig. 5):

We also analysed the distribution of differences of periodicity for each of these 3 windows in each ORF with the total periodicity of the same ORF and the majority of them do not show changes higher than 0.1 (dashed line) (now shown in Supp. Fig. 5):

d) In the section “Ribosome profiling discriminates translation abundance at isoform level” it is not clear what “combination of protein features” means in Fig 2b. It seems unsurprising that isoforms called translated have evidence of their protein expressed. It would be informative to generate a false discovery rate, for those isoforms that have no evidence of translation. It is not clear how mass spec,

immunohistochemistry and uniprot data was treated to ensure that signal could be confidently assigned to a specific ORF isoform over another?

Previous fig. 2b (now moved to Supp. Fig. 3) represents a validation at the gene level using the protein expression annotation from THPA, which is based on Mass Spectrometry, Immunohistochemistry, and Uniprot (associated known protein). The intention of this plot was to provide a first coarse-grained validation of predictions. For every gene with one or more translated ORFs predicted, we calculated whether there was evidence of translation for that gene. The plot shows that most of the genes for which we predict one more translated isoform has indeed some protein expression evidence (7992 out of 7992+365).

e) In the text, polysome association is described, whereas in Fig 2c, high polysomes are mentioned. What specific complexes were defined as high polysomes? How many transcripts were included in this monosome vs high polysome distribution? From RNA-Seq data it is not clear how we can be sure that this is transcript specific. If RT-qPCR with primers designed specifically for detection of specific isoforms match this same pattern?

Polysome fractions used in our analysis were defined in (Blair et al. Cell Reports 2017), which was based on the procedures described in (Floor & Doudna Elife 2016). Their definition was as follows: Monosomes = 1 ribosome, Low polysomes = 2-4 ribosomes, High polysomes = 5 or more ribosomes. In the table below we show the number of expressed transcripts in the analysed data:

Condition	Fraction	Expressed transcripts
hESC	High Polysome	57562
hESC	Monosome	48967
Neu	High Polysome	49342
Neu	Monosome	43023

We used Salmon to assign reads from each polysomal fractions to the transcript isoforms. Salmon performs an unambiguous assignment of reads to transcripts based on the similarities of the read to other reads mapping to the same transcript using an optimization. Abundances can be estimated to each transcript in each subpopulation of RNAs from the corresponding RNA-seq. The abundances were normalized as in (Maslon et al. Elife 2014) by dividing the abundance in a given fraction over the total abundance in all fractions being compared. In general, these normalized values are never 0 (does not appear in fraction) or 1 (unique to the fraction), but the values vary enough to determine the enrichment in specific fractions, as shown before in (Maslon et al. Elife 2014).

f) The validation performed in Fig3a is excellent. But these sections would benefit greatly from examples illustrating these types of events.

We have generated plots for isoform-specific regions with uniquely-mapped reads for a couple of examples: ENSG00000196867 and ENSG00000213995, which are now shown in Supp. Fig. 11 and Supp. Fig. 12. In these plots we show the uniquely-mapped Ribo-seq reads (green), Ribo-seq reads that map to 2 different places (red), and Ribo-seq reads that map to 3 different places (blue).

g) There is very little explanation of many of the panels. For example, 3c): what was the aim of this over how mass spec data had be used to support ORF isoform translation?

We have tried to improve the description in the legend and in the text of the various figures. In previous Fig. 3c, Mass-spec peptides from mouse hippocampus and glia were used to validate unique regions. These are two types of regions: those sequences that are specific to a single isoform (unique sequence) and those ORFs that are specific to an isoform (the sequence might be shared with other isoforms, but the ORF is unique and specific to the isoform). We have modified the cartoon in Figure 3a to clarify the possible configurations of these regions. For improved clarity, we have also separated into independent panels the validation of isoform-specific sequences with Ribo-seq reads (now in Figs. 3b and 3c) and the validation of isoform-specific ORFs with P-sites or mass-spec peptides (now in Figs. 3d and 3e).

h) It is not clear how S4 is different to Fig3? Is it same but just including human samples too?

Supp Figure 4 (now Supp. Fig. 13) represents the same type of analyses as Fig. 3 (now in Figs. 3b, 3c, 3d and 3e) but for other samples. These figures represent the density of Ribo-seq reads in isoform-specific sequences (Fig. 3b), isoform-specific sequences with 10 or more Ribo-seq reads (Fig. 3c), the density of P-sites in isoform-specific ORFs (Fig. 3d) and the number of isoform-specific ORFs with one or more peptides from Mass-Spec. Supp. Fig. 13 contains the same plots for the samples not included in Fig. 3.

i) Use of SUPPA in Fig4 to probe differential translation linked to differential splicing is really the most interesting part of the manuscript. Since one of the big questions in the field is whether certain spliced isoforms are preferentially translated. This analysis starts to address this. Fig 4c suggests this is generally not the case. The majority of analysis focuses on whether the translation of spliced microexons and their differential splicing. This is an important question and result. However, it is not clear whether this analysis was dependant on the original pipeline ORQAS, developed here.

Yes, we used ORQAS to quantify ORFs in ribosome space, and then used SUPPA to transfer that information at event level. We have tried to explain this better in the manuscript.

j) The focus of results seemed to be on changes correlating between RNA-Seq and Ribo-Seq, especially in Fig 5, that are differential between different cell types. But given the manuscript aims to understand isoform translation it would be more appropriate to analyse more deeply events whose RNA-Seq and Ribo-Seq don't correlate. These are the situations, one would argue, that require understanding of isoforms at both splicing and translation level.

Our main objective is to estimate the impact of differential splicing on translation. We argue that to be able to address that one needs to evaluate isoform translation, since differential expression and splicing is simply a result of what happens to the isoforms. Those isoforms whose RNA-seq abundance and Ribo-seq abundance do not correlate are probably related to translation efficiency (TE), which is out of the scope of this manuscript.

We considered those events that changed significantly in opposite directions in RNA and Ribosome space. We expect those that increase significantly PSI in RNA but decrease in Ribosome would be related to a decrease in TE, and those events that decrease significantly in PSI in RNA but increase in Ribosome would be related to an increase in TE. We calculated the TE in these cases:

Except for a few cases, the TE is similar in glia and glioma for all isoforms. This suggests that TE does not change globally and explains a very limited number of significant changes in splicing and translation.

k) In discussion “These estimates are far from” should be reworded to give indication of direction of change.

We have changed it to “considerably lower than”

Reviewer #3 (Remarks to the Author):

The authors have developed a new method, ORQAS (ORF quantification pipeline for alternative splicing) to quantify isoform-specific translation abundance, and have applied their method to a number of different datasets including glia/glioma and ES cells/neurons.

One of the limitations of ribosome profiling is that it does not directly measure protein peptide abundance and assumes that engaged ribosomes are direct arbiters of protein levels. As the authors point out, this is not strictly true, and builds upon previous studies to attempt to examine open reading frames that are more likely to be translated. Their polysome data is convincing.

While I have some concerns about the novelty given that other studies examining alternative splicing and ribosome profiling have previously been published, I do think that the authors have significantly improved upon these studies and this study deserves to be published in Nature Communications.

I have a few queries:

1. In Figure 4C, the correlation between the ribo-seq and RNA-seq data is remarkably high, close to 1 - almost too good to be true. What happens to the datasets that are not significant, i.e.. what is the correlation between non-significant alternative splicing events for which there is ribo-seq data and vice-versa?

We have plotted now all the events in this comparison (now shown in Fig. 4b for human and in Supp. Fig. 14c for mouse):

In these figures we show in red the cases that are significant in both Ribo-seq and RNA-seq space. In black or dark gray, we show the cases that are only significant in one case, and in light gray, the cases that are significant in neither of them. The density bands in the plot shows the distribution of cases along the axes. Empty circles indicate the microexons (red if significant, black/dark-gray/gray otherwise). In the inset in blue we give the correlation of the red points (including microexons), in gray we give the correlation of all other exons (black/dark-gray/gray, including microexons). There is overall a considerably high correlation of the non-significant cases (dark/dark-gray/gray points), but the correlation of the significant cases is higher.

2. I would suggest that the authors improve their description of their method in the first results section, including on expanding on the descriptions of uniformity and periodicity and why the integration of which is a significant advance over ribosome profiling alone.

We have added more details about the method in the first section. We have also added new analyses to justify the different steps in the method and we show a new analysis to show that the periodicity does not across ORFs. Among the tests to better explain the advantages of the ORQAS we selected genes with one single ORF annotated (different gene set from the one used as positive control in Fig. 1b) and calculated the proportion of cases that have evidence of protein expression from immunohistochemistry (IHC) experiments from the human protein atlas (THPA):

The cases labelled “Not evaluated“ are those that do not show enough RNA expression to be evaluated by ORQAS. The cases labelled “Not translated” are the predictions that Ribomap would give without any filters for periodicity or uniformity. In both cases there was an enrichment of cases without evidence of translation from THPA. In contrast, after imposing the thresholds (the cases labelled “Translated”), we observed a higher proportion of cases with protein evidence at different levels. We show this now in Figure 1c. To further show that ORQAS is superior to other methods, we have included an exhaustive comparison with SaTAnn and with the results from (Weatheritt et al. 2016). These comparisons show that ORQAS can detect more translated alternative isoforms per gene, less potential false positives, and more microexons.

ADDITIONAL REVIEWERS' COMMENTS:

Reviewer #1 (Remarks to the Author):

In our first review, our major concerns were that the paper lacked a comparison with other methods (e.g RiboMap and SaTAnn), testing of the uniformity and periodicity parameters, and additional validations such as quantification between cytoplasmic and nuclear extracts.

The authors adequately answered our concerns by:

- Testing if one single ORF genes (1005) had protein expression evidence (according to immunohistochemistry experiments from the human protein atlas), as well as being better detected by ORQAS than with RiboMap
- Comparing a Weatheritt et al 2016 dataset with ORQAS and running SaTAnn with the same samples.
- Comparing the capacity of ORQAS and SaTAnn in detecting short unique regions and microexon-containing isoforms.
- Comparing the ORF length in high-polysomal and monosomal fractions.
- Testing the robustness of ORQAS by comparing the expression of tissue-specific genes (according to THPA) and calculating translation in their glial samples.

We would have preferred if the authors had made a simulated dataset of Ribo-seq and used it to compare ORQAS, RiboMap and SaTAnn. Nevertheless, the authors nicely show that ORQAS is able to quantify transcript isoforms using Ribosome profiling reducing the number of false-positive results in comparison with previous methods.

Minor concerns:

We also asked that they should show the exact p-values in the figures and text, we request this should be done for easier interpretation. As well, we request that all figures and text should have the same consistency throughout the document showing the exact p-values and statistical tests used.

Exact p-values have been added in places where they were not present previously and the format of the figures has been unified to provide all tests as numerical p-values. The statistical test in each cases is also indicated.

We found that some minor corrections in the new data such as:

- Fig. 3f and Supp Fig. 13a the names of the x-axis are missing. We cannot tell which data is of human or mouse samples.

Figures 3f and 13a have been fixed to include the x-axis.

- In the response to referees, regarding the section discussing translation in monosomes and polysomes, the authors mention that they did not see any significant difference in the distribution of transcript lengths between monosomes and high-polysomes and show the distribution of ORF length of monosomes and high-polysomes. We ask the authors to test the homoscedasticity of these distributions, as it seems they are bimodal.

We show below the quantile-quantile (qq) plots where the quantiles of the distribution of ORF length in monosomes and high-polysomes have been plotted against theoretical quantiles of a normal distribution. The lack of interruption in the sample quantiles allows us to conclude that there is no bimodal behavior in those distributions.

Human ESC monosomes:

Human ESC high-polysome:

Human neural cells monosomes:

Human neural cells high-polysomes:

- In the methods, they should explain the specific parameters used in SaTAnn. A sentence in the methods section has been added to clarify that we used ORFquant (formerly known as SaTAnn) with default parameters and with the same ORF annotation used for ORQAS.
- All legends need to state the number of data points used in the figure. The number of data points has been added in the legend of the figures.
- The authors should also articulate the limitations of their method, for example, the use of periodicity means that events in the 5' UTR are not evaluated.

Thanks for raising this question. The main limitation of our approach is that it is based on annotated transcripts and ORFs. Accordingly, uORFs or potential new ORFs not present in the annotation are not considered. However, it can be adapted to include these cases: once novel transcripts and ORFs are predicted from RNA sequencing, one could apply ORQAS to assess their translation activity. Similarly, uORFs can be assessed based on abundance and periodicity (e.g. PMID: 31810458). So

one could envision adding those regions to analyse them with ORQAS. We have indicated these points in the Discussion section.

Reviewer #3 (Remarks to the Author):

The authors have done a thorough job addressing the reviewers concerns and the manuscript is much improved as a result. I am happy to recommend publication of this manuscript and congratulate the authors on a great study.

Thank you very much.